# Control of chrysanthemum flowering through integration with an aging pathway

Qian Wei[1], Chao Ma[1], Yanjie Xu[1], Tianle Wang[1], Yiyu Chen[1], Jing Lü[1], Lili Zhang[1], Cai-Zhong Jiang[2,3], Bo Hong[1] & Junping Gao[1]

Age, as a threshold of floral competence acquisition, prevents precocious flowering when there is insufficient biomass, and ensures flowering independent of environmental conditions; however, the underlying regulatory mechanisms are largely unknown. In this study, silencing the expression of a nuclear factor gene, *CmNF-YB8*, from the short day plant chrysanthemum (*Chrysanthemum morifolium*), results in precocious transition from juvenile to adult, as well as early flowering, regardless of day length conditions. The expression of SQUAMOSA PROMOTER BINDING-LIKE (SPL) family members, *SPL3*, *SPL5*, and *SPL9*, is upregulated in *CmNF-YB8*-RNAi plants, while expression of the microRNA, *cmo-MIR156*, is downregulated. In addition, CmNF-YB8 is shown to bind to the promoter of the *cmo-MIR156* gene. Ectopic expression of cmo-miR156, using a virus-based microRNA expression system, restores the early flowering phenotype caused by *CmNF-YB8* silencing. These results show that CmNF-YB8 influences flowering time through directly regulating the expression of *cmo-MIR156* in the aging pathway.

[1] Beijing Key Laboratory of Development and Quality Control of Ornamental Crops, Department of Ornamental Horticulture, China Agricultural University, Beijing 100193, China. [2] Crops Pathology and Genetic Research Unit, United States Department of Agriculture, Agricultural Research Service, Davis, CA 95616, USA. [3] Department of Plant Sciences, University of California at Davis, Davis, CA 95616, USA. Qian Wei and Chao Ma contributed equally to this work. Correspondence and requests for materials should be addressed to B.H. (email: hongbo1203@cau.edu.cn) or to J.G. (email: gaojp@cau.edu.cn)

Flowering at the appropriate time is critical for optimal sexual reproductive success, and is determined by a complex interplay of environmental cues and internal signals[1–3]. Angiosperms have evolved several mechanisms to coordinately regulate floral transition, including signaling controlled by photoperiod and vernalization, as well as by the gibberellin and aging pathways[2, 4]. These flowering pathways converge on a common set of downstream flowering time integrators, such as *FLOWERING LOCUS T* (*FT*), *SUPPRESSOR OF OVER-EXPRESSION OF CO1* (*SOC1*), and *APETALA1* (*AP1*), and a plant-specific transcription factor, *LEAFY* (*LFY*)[2, 5–7].

Among these flowering pathways, the aging pathway provides an endogenous developmental cue that prevents flowering during the juvenile phase, and ensures flowering during the adult phase, even in the absence of exogenous inductive factors[4, 8, 9]. A microRNA (miRNA), miR156, and its targets, *SQUAMOSA PROMOTER BINDING PROTEIN-LIKE* (*SPL*) genes, have been identified as key components of the aging pathway that underlies the transition from the juvenile-to-adult phases, and that promotes flowering[8–12]. In *Arabidopsis thaliana*, overexpression of miR156 prolongs the juvenile phase, as characterized by leaf morphology[13], whereas a reduction in miR156 levels accelerates the expression of adult traits[9, 14]. The expression of miR156 is high in seedlings, decreases with time[8], and is downregulated by the accumulation of photosynthate during vegetative growth[15, 16]. The decline in miR156 levels results in an increase in the expression of *SPL* genes, which in turn induce the expression of the floral pathway integrators *SOC1*, *AP1*, *FUL*, and *LFY* in the shoot apical meristem[4, 8, 9, 14]. A key role for miR156 in flowering has been shown to be conserved among angiosperms and ectopic expression of miR156 alters flowering time in many plant species, including rice (*Oryza sativa*)[17], maize (*Zea mays*)[18], potato (*Solanum tuberosum*)[19], *Arabis alpina*[10], and *Cardamine flexuosa*[11]. However, while some components of the complex aging pathway regulatory system have been elucidated, the upstream transcription effectors that modulate miR156 levels are still largely unknown.

Previous studies have demonstrated that flowering time, abiotic stress tolerance, and various aspects of plant development are regulated by nuclear factor Y (NF-Y) proteins[20]. NF-Y proteins comprise NF-YA, NF-YB, and NF-YC subunits that form conserved heterotrimeric transcription factor complexes in all higher eukaryotes[20]. NF-Y complexes bind to the CCAAT motif in the promoters of many genes[20, 21]. In plants, each NF-Y subunit is encoded by 8–39 genes, so there are potentially thousands of unique heterotrimeric NF-Y complexes that may enable the specific regulation of a variety of transcriptional profiles[21, 22]. As an example, *A. thaliana*, NF-YB2 and NF-YB3 bind to NF-YC3, NF-YC4, or NF-YC9 in a photoperiod-dependent flowering pathway. The heterodimers then physically interact with CONSTANS to induce the expression of *FT*[23–26]. In addition, the NF-Y complexes can function as epigenetic regulators in both the photoperiod and GA pathways[22]. However, less is known about the involvement of NF-Y proteins in the aging pathway.

Chrysanthemum (*Chrysanthemum morifolium*) is a typical obligate short day (SD) herbaceous perennial species, and has provided a useful model to study the role of day length in the transition from vegetative to inflorescence meristem identity[27]. Studies of chrysanthemum flowering to date have mainly focused

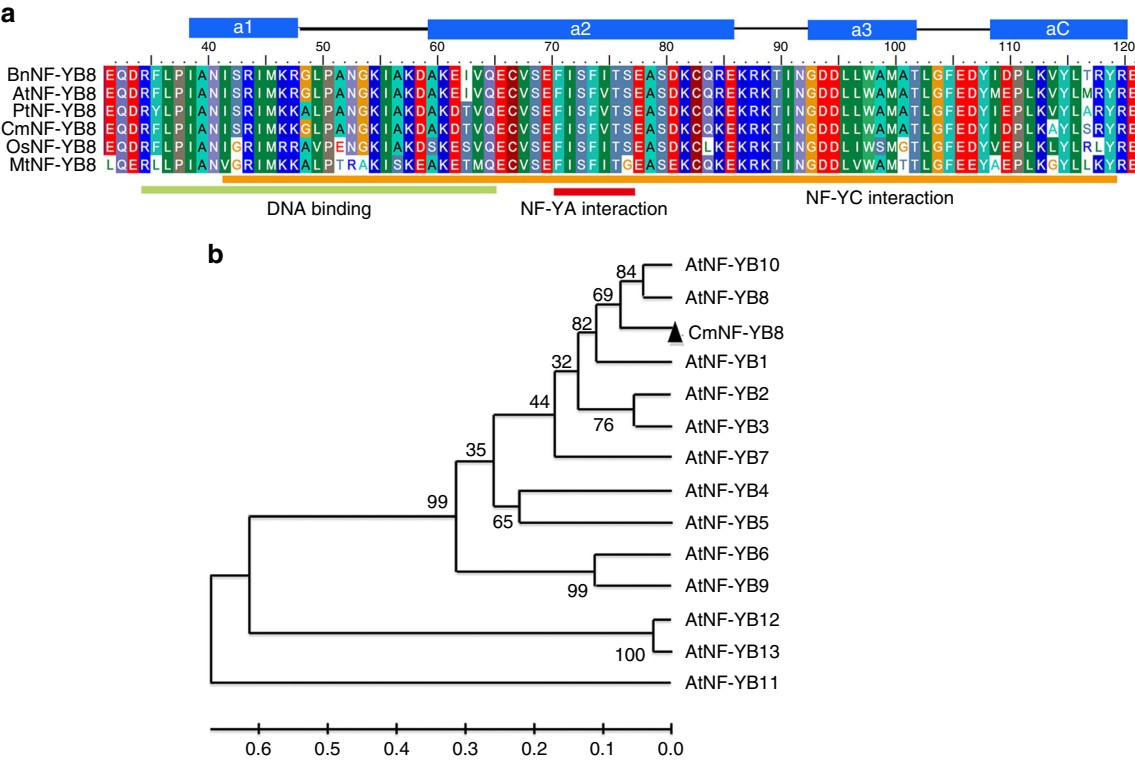

**Fig. 1** Comparison of the deduced amino acid sequence of CmNF-YB8 with NF-YB8 protein sequences from other plant species. **a** Multiple sequence alignment of CmNF-YB8 conserved domains with NF-YB proteins from other plant species. The secondary structures, alpha-helices (*solid blue rectangles*), and strand-loops (*black lines*), are represented above the alignment. The DNA-binding and subunit interaction domains are shown as *colored bars* underneath the alignment. BnNF-YB8 is from *Brassica napus*, AtNF-YB8 is from *Arabidopsis thaliana*, PtNF-YB8 is from *Populus trichocarpa*, OsNF-YB8 is from *Oryza sativa*, and MtNF-YB8 is from *Medicago truncatula*. **b** Phylogenetic analysis of CmNF-YB8 and 13 *A. thaliana* NF-YB proteins. Bootstrap values indicate the divergence of each branch, and the scale indicates branch length

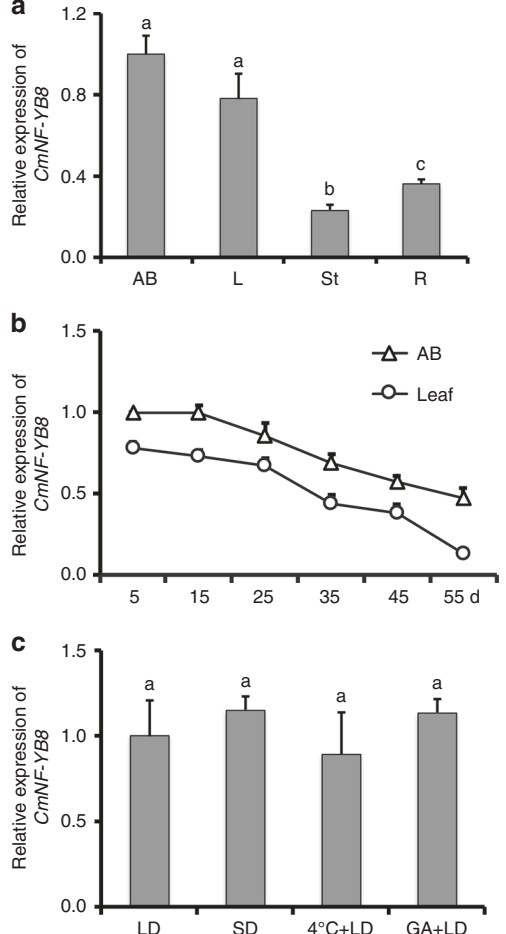

**Fig. 2** Transcript abundance of chrysanthemum *CmNF-YB8*. **a** Transcript abundance of *CmNF-YB8* in different organs. Plants at 5 days after transplanting and growth under long day (LD) conditions were used. Apical buds (AB), leaves (L), stems (St), and roots (R) were harvested. **b** Transcript abundance of *CmNF-YB8* in apical buds and leaves of differently aged chrysanthemum. Plants were grown under LD conditions. Samples were harvested every 10 days from 5 days after propagation. **c** Transcript abundance of *CmNF-YB8* in chrysanthemum grown under different day lengths conditions, including LD and SD, under a low-temperature (4 °C) regime, or following treatment with gibberellic acid (GA$_{4/7}$). Leaves were harvested 30 days after each treatment. qRT-PCR was performed to evaluate expression of *CmNF-YB8*, using *UBIQUITIN* as the control. Three independent experiments were performed and *error bars* indicate standard deviation. Significant differences were determined using Duncan's multiple range test ($P < 0.05$)

on the significance and molecular regulation of the photoperiod pathways[28–30]. It has been shown that the juvenile-to-adult transition during vegetative growth, which is required for chrysanthemum flowering, must occur before any response to SD period can take place[31]. However, little is known about the mechanism that controls the juvenile-to-adult transition in chrysanthemum. Here we report that the chrysanthemum nuclear factor CmNF-YB8 modulates flowering time by mediating the juvenile-to-adult transition. Further transcriptome analysis reveals that CmNF-YB8 regulates the expression of genes related to aging pathway. Notably, CmNF-YB8 is shown to bind to the promoter of the *cmo-MIR156* gene. Ectopic expression of cmo-miR156 restores the early flowering phenotype caused by *CmNF-YB8* silencing. Our findings uncover that CmNF-YB8 is a

critical regulator of age-dependent flowering time by directly regulating *cmo-MIR156* expression in chrysanthemum.

## Results

**CmNF-YB8 expression is associated with aging.** To investigate the function of NF-YB family genes in the aging pathway of chrysanthemum, unigenes encoding eight putative NF-YB family proteins were identified in an in-house chrysanthemum transcriptome database (Supplementary Fig. 1). Of these, UN69086, which exhibited a trend of decreasing expression in apical buds as they develop (Supplementary Fig. 2), is predicted to encode a protein (Supplementary Fig. 3) with conserved NF-YB domains, including a DNA-binding domain, and the NF-YA and NF-YC subunit interaction domains (Fig. 1a)[32, 33]. The secondary structure of the deduced UN69086 polypeptide is composed of four alpha-helices separated by three strand-loop domains, as is characteristic of the conserved domains of NF-YB family members (Fig. 1a)[34, 35]. Phylogenetic analysis revealed that the closest relationship between UN69086 and 13 *A. thaliana* NF-YB proteins was with AtNF-YB8 and AtNF-YB10 (Fig. 1b), with the greatest similarity to AtNF-YB8, so UN69086 was re-named CmNF-YB8.

The expression of *CmNF-YB8* was evaluated in different organs of young chrysanthemum plants 5 days after propagation, and transcript abundance was found to be relatively high in apical buds and leaves (Fig. 2a). In both cases, transcript levels were highest in young plants and subsequently declined during development (Fig. 2b). We also examined the expression of *CmNF-YB8* in plants grown under different day lengths, in low-temperature conditions, and following treatment with gibberellic acid (GA). No difference was observed in the expression of *CmNF-YB8* between long day (LD) and SD conditions, between plants exposed to room temperature and a low temperature, or between GA-treated or non-treated control plants (Fig. 2c). These results suggested that the expression of *CmNF-YB8* is regulated by age, but not by other known flowering signals, such as photoperiod, vernalization, or GA.

We performed a transactivation assay of CmNF-YB8 in yeast and saw no evidence that CmNF-YB8 had transcriptional activity (Supplementary Fig. 4). This is consistent with previous reports that transcriptional activation of the NF-Y heterotrimer is mediated by the NF-YA and NF-YC subunits, but not by the NF-YB subunit[21, 36].

**CmNF-YB8 silencing promotes the juvenile-to-adult transition.** To investigate whether *CmNF-YB8* influences growth and flowering time, we used the 3′ region of *CmNF-YB8* to specifically silence *CmNF-YB8* expression in chrysanthemum, generating a population of 23 *CmNF-YB8*-RNAi-expressing lines. A reduction in *CmNF-YB8* expression in the transgenic lines compared with wild-type (WT) plants was confirmed by qRT-PCR analysis (Fig. 3a). We also evaluated the expression of other NF-YB family members in the *CmNF-YB8*-RNAi lines and determined that only *CmNF-YB8* was silenced (Supplementary Fig. 5). Three representative *CmNF-YB8*-RNAi lines (4, 13, and 22) were selected for functional analysis (Fig. 3a).

We compared the morphology of leaves from the WT and *CmNF-YB8*-RNAi lines, as the number of juvenile leaves can be used as an indicator of the juvenile-to-adult transition[12, 37]. All the two first-emerging leaves and 89% of the third leaves from the WT plants were typically juvenile, in that they were small and had no, or minimal, marginal serration (Fig. 3b). In contrast, more than 20, 65, and 93% of the first- second- and third-emerged leaves, respectively, of the *CmNF-YB8*-RNAi lines had adult

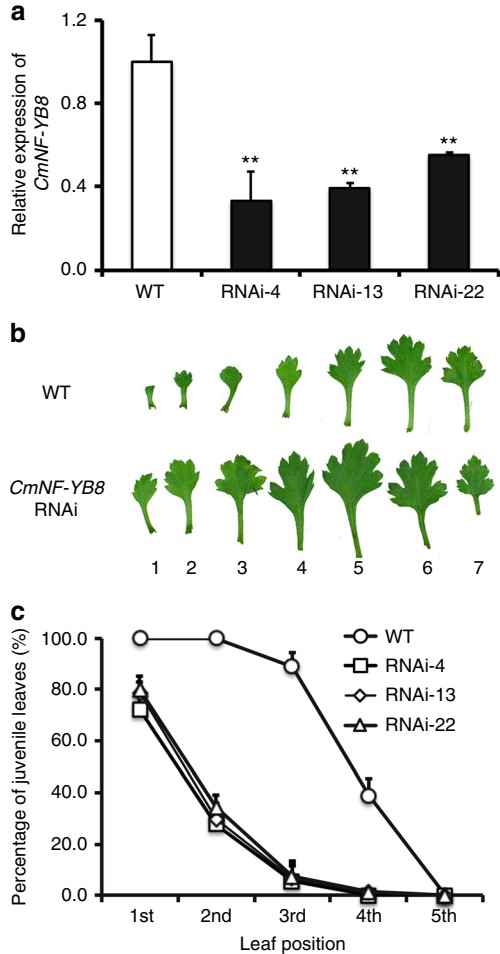

**Fig. 3** The juvenile vegetative phase of *CmNF-YB8*-RNAi chrysanthemum plants. **a** Transcript abundance of *CmNF-YB8* in wild type (WT) and *CmNF-YB8*-RNAi plants determined by qRT-PCR. RNAi-4, -13, and -22 correspond to three independent *CmNF-YB8*-RNAi lines. **b** Morphology of leaves of seven-leaf-old WT and *CmNF-YB8*-RNAi chrysanthemum plants. Juvenile leaves were defined as small, with no, or minimal, marginal serration. **c** Percentage of juvenile leaves among the first five leaves in WT and *CmNF-YB8*-RNAi chrysanthemum plants. Three independent experiments were performed and *error bars* indicate standard deviation. *Asterisks* indicate significant differences according to a Student's *t* test (**$P < 0.01$)

morphology, as they were larger and had serrated margins (Fig. 3c).

We also observed a significant difference in flowering time between *CmNF-YB8*-RNAi expressing and WT plants (Fig. 4), but no difference in flower diameter or the number of flowers. Under LD conditions, at 62 days, flower bud emergence was observed in *CmNF-YB8*-RNAi line 4, and at 66 and 68 days, lines 13 and 22 plants had flowering buds, while flower buds only emerged from the WT plants at 89 days (Fig. 4a, b). Subsequently, flower buds bloomed in *CmNF-YB8*-RNAi-expressing plants ~20 days before the WT plants (Fig. 4b). Under SD conditions, flower buds emerged at 67 days in WT plants, but after only 42 days in the *CmNF-YB8*-RNAi lines (Fig. 4c, d). Flower buds then bloomed in *CmNF-YB8*-RNAi-expressing plants ~25 days prior to the WT plants (Fig. 4d). In addition, we overexpressed (OX) *CmNF-YB8-GFP* (green fluorescent protein) in chrysanthemum, obtaining six *CmNF-YB8*-OX lines. Increased expression of *CmNF-YB8* in the transgenic lines compared to WT was

confirmed by qRT-PCR analysis (Supplementary Fig. 6a). The flowering time of *CmNF-YB8*-OX plants was slightly delayed compared with WT plants (Supplementary Fig. 6b, c).

We also tested whether *CmNF-YB8* has the capacity to regulate juvenile-to-adult leaf transition and flowering time in *A. thaliana*. *CmNF-YB8* was overexpressed in *A. thaliana*, and three of the resulting 15 *CmNF-YB8*-OX lines (OX7, OX8, and OX11) were chosen for functional analysis (Supplementary Fig. 7a). In *A. thaliana*, a marker for juvenile leaves is no, or few, trichomes on the abaxial side[14]. Compared to WT plants, the *CmNF-YB8*-OX plants had a prolonged juvenile phase since the appearance of abaxial trichomes was delayed by more than 1.3 plastochrons (Supplementary Fig. 7c, d). Furthermore, *CmNF-YB8*-OX *A. thaliana* lines flowered later than WT, as evidenced by the fact that, at the bolting stage, the transgenic plants had more rosette leaves than did the WT plants (Supplementary Fig. 7b, c).

Taken together, these results suggested that the influence of CmNF-YB8 on flowering time might be exerted through the aging pathway.

**CmNF-YB8 regulates the expression of aging pathway genes.** We performed a large-scale screen of genes that were differentially expressed between the aerial organs of *CmNF-YB8*-RNAi and WT chrysanthemum plants using an RNA-sequencing (RNA-seq) approach. In total, we identified 683 upregulated and 74 downregulated differentially transcribed genes (DTGs) in *CmNF-YB8*-RNAi line 4 (Supplementary Data 1). In the context of the regulation of flowering time by CmNF-YB8, we focused on DTGs that were annotated as components of flowering pathways. These included three putative aging pathway genes, encoding SPL proteins, which were expressed at higher levels in the *CmNF-YB8*-RNAi lines than in WT plants (Table 1). Based on an alignment with, and sequence homology to, *A. thaliana* SPL proteins (Supplementary Fig. 8), we designated these three genes as *CmSPL3*, *CmSPL5*, and *CmSPL9* (Table 1). In addition, DTGs encoding the known chrysanthemum flowering integrators *CmFTL3* and *CsAPETALA1/FRUITFULL*[28, 29] were also expressed at higher levels in the transgenic lines (Table 1). To validate the RNA-seq data, we measured the transcript abundance of the *CmSPL* genes in the *CmNF-YB8*-RNAi chrysanthemum by qRT-PCR and observed that the patterns of expression indicated by the two approaches were generally consistent (Fig. 5a–c).

Since miR156 is a critical component of the aging pathway, we first identified a miR156 from chrysanthemum apical bud RNA-seq data, named cmo-miR156, and determined the sequence of the *cmo-MIR156* gene from the chrysanthemum genome sequence. qRT-PCR analysis showed that both the expression of cmo-miR156 and the primary transcript of cmo-MIR156 (pri-*cmo-MIR156*) were downregulated in apical buds as a consequence of *CmNF-YB8* silencing (Fig. 5d, e).

To confirm the regulation of *CmSPL3*, -5, and -9 transcripts by cmo-miR156, we identified the cleavage products of the three *CmSPL* genes in RNA extracts from chrysanthemum apical buds by 5′ RNA ligase-mediated rapid amplification of cDNA ends (RLM)-RACE. As shown in Fig. 6a, the cmo-miR156 cleavage site in *CmSPL3* was between nucleotides 9 and 10 from the 5′ end of cmo-miR156, and the cleavage site on *CmSPL9* was between nucleotides 10 and 11 (Fig. 6a). No cmo-miR156 cleavage site was identified in *CmSPL5*, although it has a putative cmo-miR156-binding site (Fig. 6a).

We also performed a dual-luciferase-based miRNA sensor assay to quantitatively evaluate the cleavage of the *CmSPL* genes by cmo-miR156 in a *Nicotiana benthamiana* heterologous expression system. The *cmo-MIR156* gene has two local

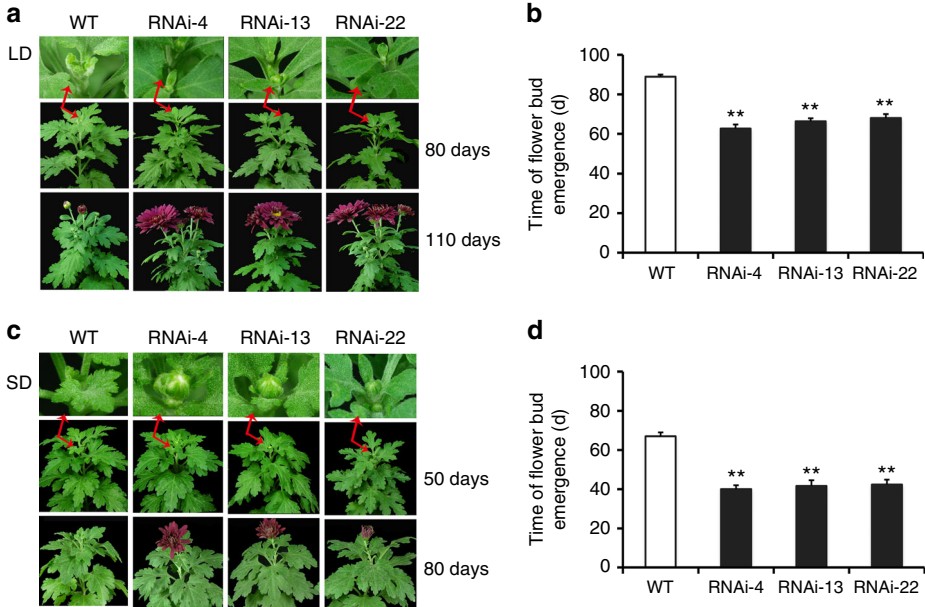

**Fig. 4** Flowering time of *CmNF-YB8*-RNAi chrysanthemum plants under long day and short day conditions. **a, c** The generative phenotypes of wild type (WT) and RNAi plants, grown under long day (LD) (**a**) and short day (SD) (**c**) conditions, were recorded and photographed at different time points (days after transplanting). **b, d** Flower bud emergence time in WT and RNAi plants grown under LD (**b**) or SD (**d**) conditions. Three independent experiments were performed and *error bars* indicate standard deviation. *Asterisks* indicate significant differences according to a Student's *t* test (**$P < 0.01$)

**Table 1 Genes related to flowering time with upregulated expression in *CmNF-YB8*-RNAi-expressing chrysanthemum**

| Gene | Annotation | WT[a] | RNAi[a] | RNAi/WT fold change |
|---|---|---|---|---|
| *Aging related* | | | | |
| UN052587 | *CmSPL3* | 14.18 | 28.61 | 2.02 |
| UN011609 | *CmSPL5* | 0.60 | 1.50 | 2.50 |
| UN033721 | *CmSPL9* | 21.77 | 42.36 | 2.08 |
| *Flowering integrators* | | | | |
| UN043322 | *CmTFL1-like protein*[b] | 0 | 4.85 | CNC |
| UN062942 | *CmFTL3*[b] | 5.87 | 10.97 | 1.87 |
| UN074378 | *SOC1* | 7.28 | 11.26 | 1.55 |
| UN055907 | *SOC1-like protein 1* | 9.09 | 17.29 | 1.90 |
| UN026081 | *CsAPETALA1/ FRUITFULL*[b] | 5.64 | 25.50 | 4.52 |
| UN045503 | *APETALA2-like protein* | 2.29 | 5.59 | 2.45 |

CNC indicates cannot calculate. WT, wild type
[a]Data in the columns are RPKM values
[b]reported as flowering integrators in chrysanthemum[28, 29]

stem-loop structures, named precursor cmo-miR156a (pre-cmo-miR156a, 101 bp) and pre-cmo-miR156b (122 bp) (Figs. 6b and 7a). The potential cmo-miR156 cleavage sites are located at the 3′ UTR of *CmSPL3* and *5*, and in the open-reading frame (ORF) of *CmSPL9*. Co-expression in *N. benthamiana* of pre-cmo-miR156a (35S:pre-cmo-miR156a) with either of the *CmSPL3* cleavage sites fused to the 3′UTR of the *Firefly luciferase* (*LUC*) gene or the *CmSPL9* cleavage sites fused to the ORF of the *LUC* gene led to reduced LUC activity. However, this was not observed when pre-cmo-miR156a was co-expressed with the *CmSPL5* cleavage site fused to the 3′UTR of the *LUC* gene (Fig. 6c).

**CmNF-YB8 is an upstream regulator of *cmo-MIR156*.** To determine whether CmNF-YB8 is associated with specific regulatory regions of the *cmo-MIR156* promoter, we performed a chromatin immunoprecipitation (ChIP) assay using CmNF-YB8-GFP overexpressing chrysanthemum. We observed that the association of CmNF-YB8 with the *cmo-MIR156* promoter had an enrichment peak in the P5 fragment, $-1121 \sim -1318$ bp relative to a major transcription start site (TSS), but not with other fragments of the *cmo-MIR156* promoter, or with a pre-cmo-miR156a (P0) control fragment (Fig. 7a, b).

It has been reported that the DNA-binding property of NF-YB can be evaluated in yeast, since the yeast HAP complex members, HAP2 and HAP5, can act as surrogate NF-YA and NF-YC partners, and combine with NF-YB, thereby forming a functional NF-Y complex[24]. Using a yeast one-hybrid assay, we observed that CmNF-YB8 bound to the P9 fragment ($-1083 \sim -1436$ bp relative to the TSS) of the *cmo-MIR156* promoter (Fig. 7d). A putative CCAAT box *cis*-element ($-1217 \sim -1221$ bp), which is a known binding site for NF-Y proteins[21], was localized at the overlapping region of P5 and P9 (Fig. 7a). The binding activity was completely abolished by mutation of the CCAAT box (Fig. 7c, d). In addition, we performed a dual-luciferase reporter assay to analyze the regulation by CmNF-YB8 of the promoter activity of *cmo-MIR156* in vivo (Fig. 7e). *N. benthamiana* leaf cells co-transformed with *35S:CmNF-YB8* (CmNF-YB8) and promoter *cmo-MIR156:LUC* (Pro-miR156) had significantly higher LUC activity than cells transformed with CmNF-YB8 or Pro-miR156 alone, or with Pro-miR156 carrying a mutated CAAT *cis*-element (Fig. 7e).

Finally, we evaluated whether CmNF-YB8 can bind to the *MIR156* promoters in *A. thaliana*, as was suggested by the observation that the expression of *SPL3, 5, 9* was lower, and miR156 abundance was higher in the *CmNF-YB8*-OX *A. thaliana* lines (Supplementary Fig. 9). A ChIP assay using the *CmNF-YB8-GFP*-OX *A. thaliana* lines demonstrated that CmNF-YB8 associated with *MIR156A* in a promoter region $-824 \sim -351$ bp relative to the TSS of *MIR156A*, and in *MIR156C* in a region $-759 \sim -446$ bp relative to the major TSS of *MIR156C*[38] (Supplementary Fig. 10).

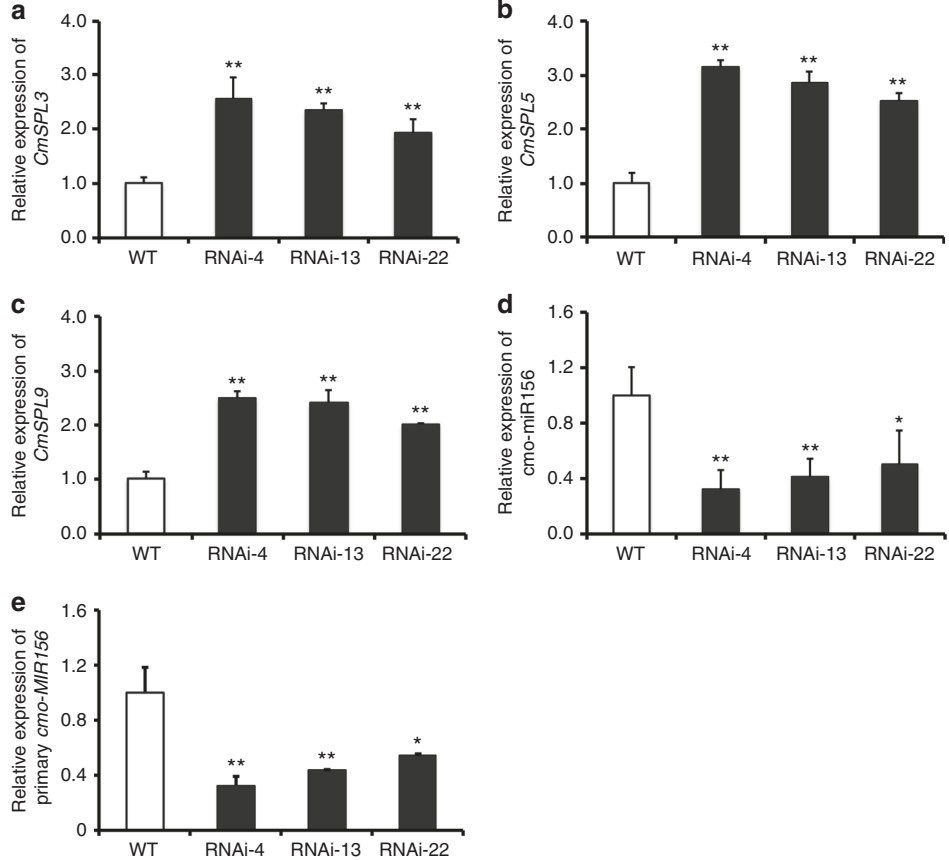

**Fig. 5** Expression of *CmSPL* genes (**a**, **b**, **c**) and *cmo-MIR156* (**d**, **e**) in *CmNF-YB8*-RNAi chrysanthemum plants. qRT-PCR was performed to evaluate the expression of each gene. *UBIQUITIN* was used as the control gene for *CmSPL3*, *CmSPL5*, *CmSPL9*, and primary *cmo-MIR156* expression. *U6* was used as the control gene for cmo-miR156. Three independent experiments were performed and *error bars* indicate standard deviation. *Asterisks* indicate significant differences according to a Student's *t* test (*$P < 0.05$, **$P < 0.01$)

**cmo-miR156 restores early flowering by *CmNF-YB8* silencing.** To gain additional evidence that *CmNF-YB8* influences flowering time through a cmo-miR156-SPL module, we overexpressed cmo-miR156 in WT and *CmNF-YB8*-RNAi chrysanthemum lines, using a modified cabbage leaf-curl geminivirus vector (CaLCuV) containing pre-cmo-miR156a (CaLCuV + miR156)[39]. As expected, in both WT and RNAi plants, expression of cmo-miR156 was increased in CaLCuV + miR156-infected plants compared to the Mock and CaLCuV-only infected control plants (Fig. 8a). Under SD conditions, at 50 days, scanning electron microscopic observation showed that in both WT and RNAi plants, inflorescence development was slower in CaLCuV + miR156-infected plants than in control plants (Fig. 8b). In RNAi plants, CaLCuV + miR156 infection restored inflorescence development to the stage of involucre formation; the same stage as the WT control plants (Fig. 8b). Subsequently, following CaLCuV + miR156 infection, flower bud emergence occurred on day 98 and 77 in WT and RNAi plants, respectively, which represented a significant delay compared to the control plants (Fig. 8c, d). In addition, while we noted a 10 days difference in flowering time between the RNAi and the WT control plants as a consequence of CaLCuV + miR156 infection, in the context of this experiment this was not statistically significant (Fig. 8d).

## Discussion

Plants undergo a transition from a vegetative to a reproductive developmental phase after germination, the former of which can be further divided into juvenile and adult stages[40]. These two stages can be distinguished morphologically by leaf shape, size, and patterns of epidermal differentiation[41]. For example, the juvenile leaves of *A. thaliana* are smaller and rounder than the adult leaves, and they lack abaxial trichomes[14]. In the present study, we noted that juvenile leaves of chrysanthemum have similar morphological characteristics to those of *A. thaliana* (Fig. 3b), although no equivalent differences were observed with respect to abaxial trichome distribution or the structures of the juvenile and adult leaves.

Plants acquire the ability to flower and undergo the transition to reproductive development during the transition from juvenile-to-adult[42], and plants in the juvenile phase are thought to have no flowering competence, and will remain vegetative even when exposed to flowering cues such as photoperiod or vernalization. The propensity of plants to flower and to initiate reproduction increases with age[43], so the control of age-related competence to flower should be distinct from competence mechanisms regulated by environmental responses such as photoperiod, or by general endogenous pathways, such as the GA pathway[43]. First, the expression of genes in the aging-related pathway should be affected by age, rather than by other flowering cues[8]. In several plant species, such as *A. thaliana*[8, 14], *A. alpina*[10], *Acacia confusa*[12], *Eucalyptus globulus*[12], *Hedera helix*[12], *Z. mays*[44], *O. sativa*[45], and *Glycine max*[46], the *miR156/SPL* module shows a typical temporally regulated expression pattern, regardless of other flowering cues. In young *A. thaliana* seedlings, the expression levels of miR156 are high, and decline with growth, a trend that appears to be independent of known floral regulators, including vernalization, GA, and photoperiod[8]. Here we found that the expression of *CmNF-YB8* in chrysanthemum

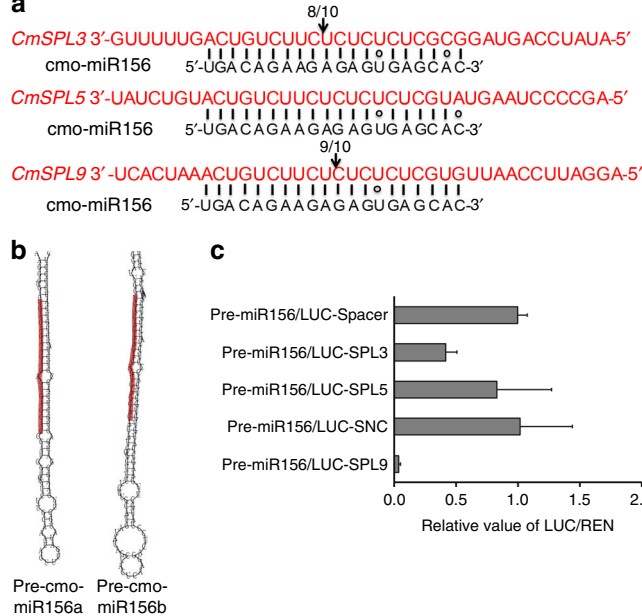

**Fig. 6** cmo-miR156 targets *CmSPL3* and *CmSPL9*. **a** cmo-miR156-directed cleavage sites in the *CmSPL* genes were determined by 5′ RLM-RACE, and are highlighted by *arrows*. The number indicates the frequency of clones when validating the cleavage sites of the target mRNAs. **b** The precursor structures of cmo-miR156 were predicted by RNA structure software. miRNA sequences are highlighted in *red*. **c** A dual-luciferase reporter assay was used to quantitate cmo-miR156-mediated repression of target gene *CmSPLs*. 35S:cmo-miR156 precursor a (pre-miR156a) was co-transformed with *CmSPL3* or *5* cleavage sites fused to the 3′UTR of the *LUC* gene, *CmSPL9* cleavage site fused to the ORF of the *LUC* gene. LUC vectors carry a *REN* gene under the control of the 35S promoter as a positive control. Synonymous negative control (SNC) is a negative control with disruption of *CmSPL9* cleavage site while maintaining identical amino acid sequence. Spacer is a 20 nt random sequence as negative control for *CmSPL3* and *CmSPL5*. *Nicotiana benthamiana* leaves were infiltrated with the samples, and LUC and REN activities were assayed 3 days after infiltration. Three independent experiments were performed and *error bars* indicate standard deviation

was regulated by age, but not by day length, low temperature, or GA (Fig. 2). Second, the abundance of aging-related gene transcripts should depend on the age of the shoot to modulate the transition from the juvenile to the adult vegetative phase. In *A. thaliana*, silencing of miR156, or overexpression of *SPL* genes, shortens the juvenile phase and causes a decrease in the number of juvenile leaves[8, 14, 47, 48]. In this current study, silencing of a NF-YB subunit member, *CmNF-YB8*, in chrysanthemum resulted in a significant reduction in the number of juvenile leaves, indicating a precocious transition from juvenile to adult (Fig. 3b, c), with a concomitant acceleration in the transition to flowering under both LD and SD conditions (Fig. 4). These observations suggest that CmNF-YB8 functions in regulating flowering time predominately through the aging pathway.

It has been reported that miR156 plays a critical role in the aging pathway by directly regulating SPL transcription factors at a posttranscriptional level[48, 49]. Here we showed that the major cmo-miR156 cleavage sites in *CmSPL* members were different (Fig. 6a). Although the cleavage site in *CmSPL9* was located between the canonical tenth and eleventh nucleotides, the site in *CmSPL3* was between the ninth and tenth nucleotide (Fig. 6a). In addition, although no site was identified in *CmSPL5* (Fig. 6a, c), the expression of *CmSPL5* was increased by *CmNF-YB8*

silencing (Fig. 5b), implying that other miR156 variants, such as other family members or iso-miR156s, might contribute to the cleavage.

In this study, we found that CmNF-YB8 regulates the transcription of the *cmo-MIR156* gene by associating with its promoter (Fig. 7). ChIP-PCR and yeast one-hybrid assays indicated that CmNF-YB8 binds to the promoter of *cmo-MIR156* (Fig. 7a–d). Interestingly, dual-luciferase reporter assay showed that in *N. benthamiana* leaf cells, the promoter activity of *cmo-MIR156* could be induced by overexpression of *CmNF-YB8* (Fig. 7e), although CmNF-YB8 has no transcriptional activity (Supplementary Fig. 4). It is well known that the regulation of downstream gene by NF-YB subunit need interact with NF-YA and NF-YC subunits to form heterotrimeric complex[20, 21]. Therefore, we speculate that CmNF-YB8 can induce the promoter activity of *cmo-MIR156* through recruiting NF-YA and NF-YC of *N. benthamiana* to form a functional NF-Y complex. In *A. thaliana*, the AGAMOUS-like (AGL) transcription factors, AGL15 and AGL18, act as upstream regulators of *MIR156* expression by forming a heterodimer, and binding to the CArG motifs of the *MIR156A* and *MIR156C* promoters[50]. Transgenic *A. thaliana* plants overexpressing *AGL15* or *AGL18* had shortened petioles and wavy/curled leaves, rather than abnormal juvenile leaf numbers[51]. *AGL15* or *AGL18* can therefore be considered to function in a complex, along with other components to regulate miR156 expression. Further studies are required to investigate whether NF-YB8 associates with AGL15/18 in the aging pathway.

In *A. thaliana*, genes of NF-YB subunit have been reported as regulators of flowering time in photoperiod and GA pathways. NF-YB2 was discovered and it can physically associate with the CCAAT box of the *FT* promoter in photoperiod-dependent flowering pathway[23, 24, 26]. Subsequently, it was found that NF-YB2 directly regulates *SOC1* expression in response to photoperiod and GA flowering pathways by interacting with NF-YA2 and NF-YC9[22]. Here we found a new function of NF-YB subunit members in flowering time. CmNF-YB8 regulates the transition of juvenile-to-adult through directly targeting *cmo-MIR156* in chrysanthemum. It suggests broad-spectrum functions of NF-YB subunit members in regulation of flowering time in response to environmental cues and internal signals.

In addition, the NF-Y complex has been demonstrated to be an epigenetic regulator that associates with *SOC1* promoter to mediate H3K27me3 demethylation, partly through a H3K27 demethylase REF6 in both the photoperiod and GA flowering pathways[22]. It is known that during the vegetative phase change, epigenetic regulation of miR156 expression is mediated by a balance between H3K27ac and H3K27me3 methylation levels[52, 53]. Further investigation to determine more precisely how NF-YB8 regulates miR156 expression will provide a better mechanistic understanding of how aging controls the juvenile-to-adult transition. We conclude that CmNF-YB8 is a regulator that coordinates the perception of aging signals, and directly triggers miR156-SPL-regulated molecular processes, thereby modulating chrysanthemum flowering time (Fig. 9).

## Methods

**Plant materials and treatments**. A ground-cover chrysanthemum cultivar (*Chrysanthemum morifolium* cv. Fall Color) used in this study was propagated by in vitro culturing. Chrysanthemum shoots, including 1 node, were cultured on medium comprising 1/2 Murashige and Skoog (MS) for 40 days, then transplanted into 9 cm diameter pots containing a mixture of 1:1 (v/v) peat and vermiculite and grown in a culture room at $23 \pm 1$ °C, 40% relative humidity, and 100 µmol m$^{-2}$ s$^{-1}$ illumination with fluorescent lamps under a LD cycle (16 h light/8 h dark).

For SD treatments, plants grown for 5 days after transplanting were transferred to a culture room and maintained under an SD cycle (8 h light/16 h dark). For low-temperature treatments, plants were transferred to a 4 °C chamber under a LD

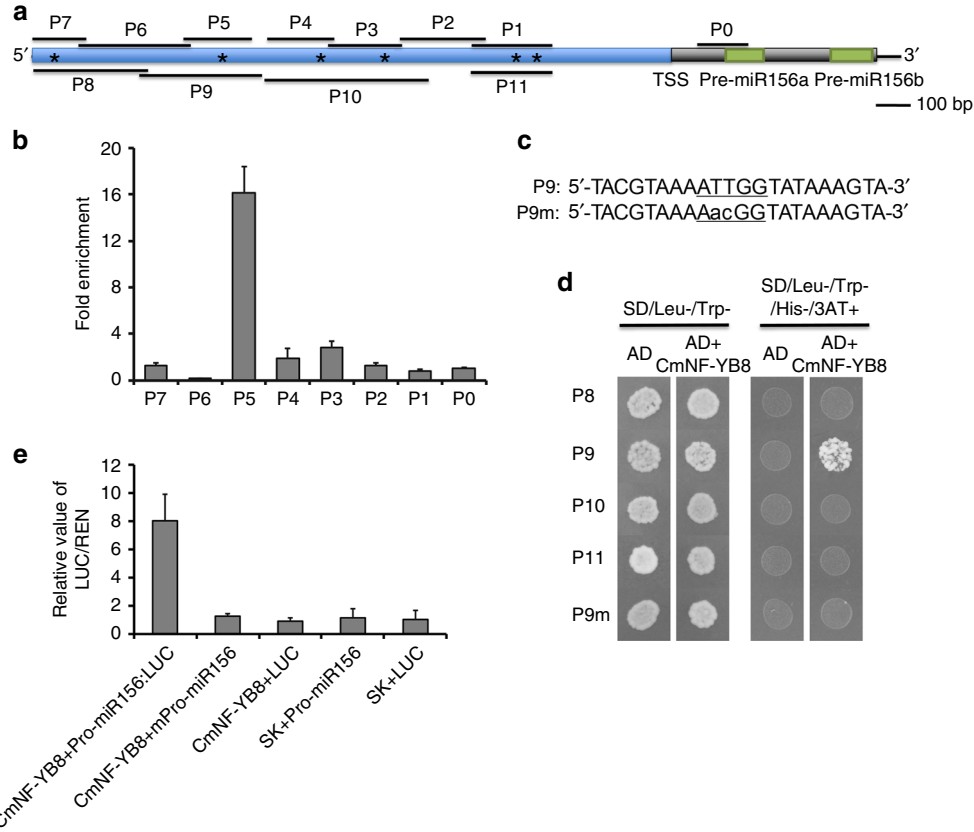

**Fig. 7** CmNF-YB8 binds to the promoter of *cmo-MIRNA156*. **a** Schematic representation of the upstream region of the foldback structure of the cmo-miR156 precursors a and b (pre-miR156a and pre-miR156b, respectively; *green boxes*). The *blue box* indicates the promoter of the *cmo-MIR156* gene. *Asterisks* correspond to putative CCAAT boxes and *lines* above the *blue box* indicate the fragments amplified in the ChIP-PCR analysis. P0: +175 ~ +321 bp, P1: −266 ~ −498 bp, P2: −473 ~ −720 bp, P3: −701 ~ −914 bp, P4: −893 ~ −1086 bp, P5: −1121 ~ −1318 bp, P6: −1289 ~ −1621 bp, P7: −1599 ~ −1754 bp relative to the major transcription start site (TSS) of *cmo-MIR156*. *Lines* below the *blue box* indicate the fragments used in the yeast one-hybrid analysis shown in **d**. P8: −1421 ~ −1754 bp, P9: −1083 ~ −1436 bp, P10: −610 ~ −1082 bp, P11: −263 ~ −501 bp. **b** ChIP-enrichment of the indicated fragments (P0–P7) in the *cmo-MIR156* promoter. Chromatin from *35S:CmNF-YB8-GFP* chrysanthemum was immuno-precipitated with an anti-GFP antibody, and the amount of the indicated DNA was determined by qPCR. A pre-cmo-miR156a fragment (P0) was amplified as a negative control. **c** The P9 wild-type *cis*-element (CAAT box) and its nucleotide substitutions in the mutant version (P9m) are underlined. **d** Analysis of CmNF-YB8 binding to the promoter of *cmo-MIR156* in a yeast one-hybrid system. The empty prey vector (AD) was used as a negative control. Interactions between bait and prey were determined by cell growth on synthetic dropout nutrient medium lacking Trp, Leu, and His, and containing 3 mM 3-amino-1,2,4-triazole (3-AT). **e** Interaction between CmNF-YB8 and *cmo-MIR156* promoter using a dual-luciferase reporter assay in *Nicotiana benthamiana* leaves. A *cmo-MIR156* promoter fragment: 0 ~ −1436 bp (from P9 to TSS) was used in this assay. mPro-miR156 is the same promoter fragment with a mutated CAAT *cis*-element (P9m). LUC vectors contain the *REN* gene under the control of the 35S promoter as a positive control. Samples were infiltrated into *N. benthamiana* leaves, and LUC and REN activities were assayed 3 days after infiltration. The ratio of LUC/REN of the empty vector (SK) co-transformed with LUC empty vector was used as calibrator (set as 1). Three independent experiments were performed and *error bars* indicate standard deviation

cycle (16 h light/8 h dark) for 30 days. For GA treatments, plants at 5 days after transplanting were sprayed with 100 μM GA$_{4/7}$ (Sigma) and maintained under a LD cycle (16 h light/8 h dark). The GA$_{4/7}$ treatment was performed twice every week for 30 days[29].

**Gene isolation**. A partial *CmNF-YB8* sequence was obtained from a chrysanthemum RNA-seq data set, and the full-length sequence was obtained using a SMART™ RACE cDNA amplification kit (Clontech), according to the manufacturer's instructions.

Alignments of the deduced CmNF-YB8 full-length amino acid sequence with *A. thaliana* NF-YB sequences, and of the deduced CmSPLs amino acid sequence with *A. thaliana* SPL sequences were performed using BioEdit (http://www.mbio.ncsu.edu/BioEdit/bioedit.html) ClustalW (http://www.ch.embnet.org/software/ClustalW.html). Phylogenetic analyses were performed using MEGA version 5 and the neighbor-joining method with 1000 bootstrap replicates.

The full-length *cmo-MIR156* primary transcript was cloned using the FirstChoice® RLM-RACE kit (Ambion)[38]. For 5′ RLM-RACE, 1 μg total RNA was directly ligated to the 5′ adapter using the T4 RNA ligase treated with calf intestine alkaline phosphatase and tobacco acid pyrophosphatase. The ligated RNA sample was used to synthesize cDNA using M-MLV reverse transcriptase. *cmo-MIR156* gene-specific primers were designed based on sequence immediately upstream of the predicted foldback structure. 3′ RLM-RACE was performed according to the

manufacturer's instructions (Ambion). PCR primers are listed in Supplementary Table 1.

The genome sequence of *cmo-MIR156* was obtained from in-house unpublished chrysanthemum genome sequencing data. AtNF-YB8 (AT2G37060) sequences were obtained from the TAIR database (www.arabidopsis.org) and OsNF-YB8 (AB095440); BnNF-YB8 (AHI94926) and MtNF-YB8 (AFK49658) sequences were obtained from NCBI (http://www.ncbi.nlm.nih.gov/); PtNF-YB8 (Potri.008G044800.1) sequences were obtained from PopGenIE (http://popgenie.org/). *A. thaliana* NF-YB family sequences were obtained from the TAIR database: AtNF-YB1 (AT2G38880), AtNF-YB2 (AT5G47640), AtNF-YB3 (AT4G14540), AtNF-YB4 (AT1G09030), AtNF-YB5 (AT2G47810), AtNF-YB6 (AT5G47670), AtNF-YB7 (AT2G13570), AtNF-YB8 (AT2G37060), AtNF-YB9 (AT1G21970), AtNF-YB10 (AT3G53340), AtNF-YB11 (AT2G27470), AtNF-YB12 (AT5G08190), AtNF-YB13 (AT5G23090). A partial sequence of CmNF-YB8 has been deposited as NFYB7 (KT253142) in the NCBI database; however, based on our phylogenetic analysis, we re-named it as CmNF-YB8.

**Quantitative RT-PCR analysis**. Total RNA samples were extracted from chrysanthemum apical buds, leaves, stem, and roots using TRIzol reagent (Invitrogen) and treated with RNase-free DNase I (Promega). First-strand cDNAs were synthesized from 1 μg total RNA using an oligo d(T) primer and the SuperScript III RT-PCR system, according to the manufacturer's instructions

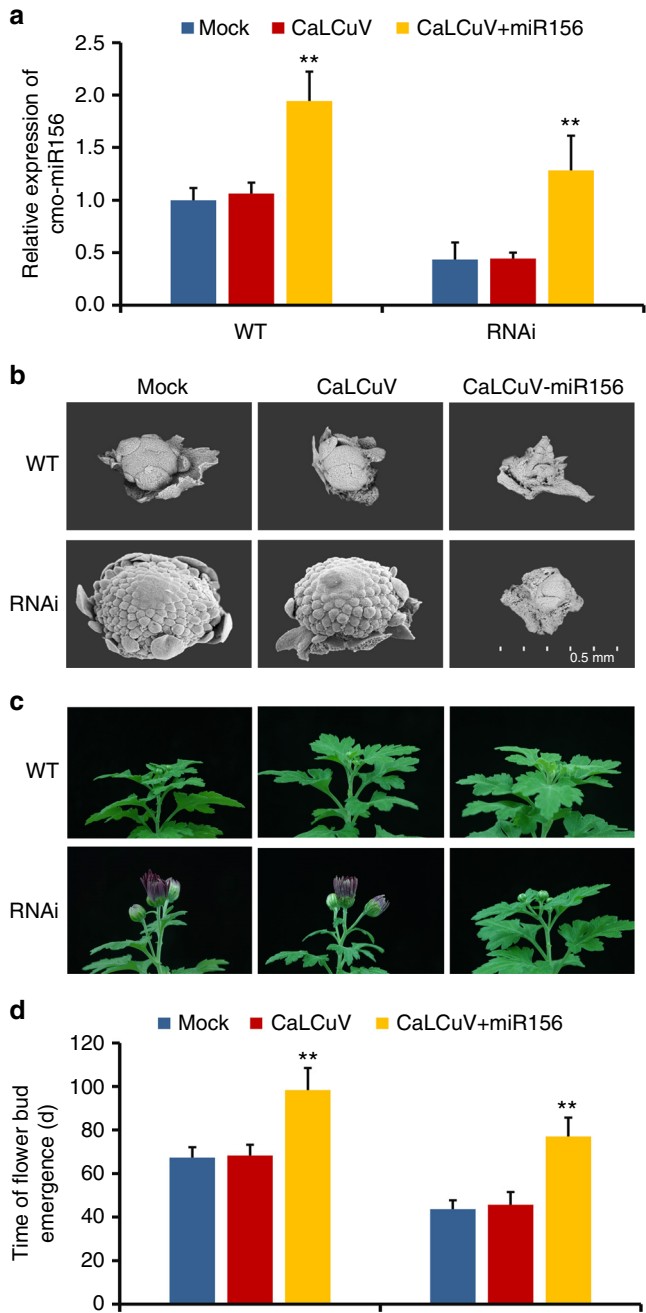

**Fig. 8** Flowering time is restored by ectopic expression of cmo-miR156 in *CmNF-YB8*-RNAi expressing chrysanthemum lines. **a** Transcript abundance of cmo-miR156 in wild type (WT) and *CmNF-YB8*-RNAi expressing plants infected with CaLCuV, or CaLCuV-pre-cmo-miR156a (CaLCuV + miR156). **b** Scanning electron micrographs of the inflorescence after 50 days of growth under short day (SD) conditions. **c** The generative phenotypes of WT and RNAi plants were recorded after 65 days of growth under SD conditions. **d** The time of flower bud emergence in WT and *CmNF-YB8*-RNAi expressing plants infected with CaLCuV or CaLCuV + miR156 grown under SD conditions. Three independent experiments were performed and *error bars* indicate standard deviation. *Asterisks* indicate significant differences according to a Student's *t* test (**$P < 0.01$)

(Invitrogen). qRT-PCR reactions (20 μl volume containing 1 μl cDNA as the template) were run using the StepOne Real-Time PCR System (Applied Biosystems) in standard mode with the KAPA SYBR FAST Universal qRT-PCR Kit (Kapa Biosystems). The chrysanthemum *UBIQUITIN* gene (GenBank accession NM_112764) was used as an internal control.

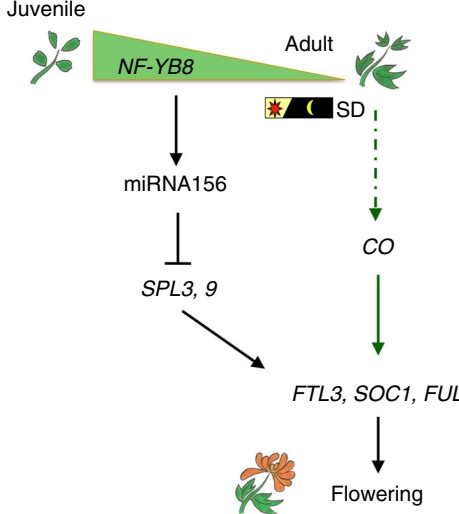

**Fig. 9** Schematic model describing the involvement of *CmNF-YB8* in the chrysanthemum aging pathway

Stem-loop reverse transcription[54] was performed to evaluate the expression of cmo-miR156. cDNAs were synthesized using a miR156 stem-loop primer and the SuperScript III RT-PCR system as described above. qRT-PCR reactions (20 μl volume containing 1 μl cDNA and the cmo-miR156F/stem-loop universal-R primer set) were run using the StepOne Real-Time PCR System (Applied Biosystems) as described above. The *U6* gene was used as an internal control[55] for stem-loop qRT-PCR.

Each reaction was performed using three biological replicates and verified by melting curve analysis. All reactions were performed with at least three biological replicates. PCR primers are listed in Supplementary Table 1.

**Chrysanthemum transformation.** To construct the RNAi vector, 312 bp sense and antisense *CmNF-YB8* fragments were amplified by two pairs of primers containing either *Xho*I/*Kpn*I or *Xba*I/*Bam*HI sites. The PCR products were digested with the corresponding restriction enzymes, and directionally inserted into the two sides of the Pdk Intron in the pHANNIBAL vector to form an intron-containing "hairpin" RNA (ihpRNA) construct. The ihpRNA construct, containing a constitutive 35S promoter and a Nos terminator, was then subcloned into the pART27 binary vector for transformation[56].

To construct the *35S:CmNF-YB8-GFP* overexpressing vector, the *CmNF-YB8* ORF fused to *GFP* was digested with *Xba*I and *Sma*I from the *Super:CmNF-YB8-GFP* vector, and then cloned into the pBI121 vector. PCR primers used for vector construction are listed in Supplementary Table 1.

The resulting plasmids were introduced into *Agrobacterium tumefaciens* strain EHA105 and transformed into chrysanthemum by *Agrobacterium*-mediated transformation[57]. The kanamycin-resistant primary transformants were screened by PCR to confirm whether the plants contained the transgene.

**A. thaliana transformation.** To construct the *CmNF-YB8* overexpression vector, the ORF of *CmNF-YB8* was amplified by a pair of primers containing *Pst*I and *Sal*I restriction sites. The PCR products were digested with the corresponding restriction enzymes, and inserted into the Super1300 vector at the 5′ end of the sequence encoding GFP. The PCR primers are listed in Supplementary Table 1. The *Super:CmNF-YB8-GFP* plasmid was introduced into *A. tumefaciens* strain GV3101, which was then used to transform *A. thaliana* using the floral dip method[58]. Independent transformants were screened on MS basal medium containing 50 mg l⁻¹ kanamycin. T3 plants were used in this study.

**Trichome staining.** Whole rosettes were washed in acetic acid:ethanol (1:3) for 3 h, and then washed three times in deionized water. The samples were incubated overnight in aniline blue solution (150 mM KH₂PO₄ and 0.01% (w/v) aniline blue, pH 9.5). The autofluorescence of callose in the trichomes was observed using a 405-nm laser with an emission window of 490–510 nm[59].

**RNA-seq analysis.** Total RNA samples were extracted from the aerial parts of *CmNF-YB8*-RNAi line 4 and WT plants (three independent plants of each genotype) 15 days after transplanting, using the RNeasy Plant Mini Kit (QIAGEN). Six RNA-Seq libraries were prepared[60] and sequenced using a HiSeq 2000 (Illumina) at the RIBO Biotechnology Co. Ltd (www.ribobio.com). RNA-seq data were processed, assembled, and annotated as previously described[29]. Briefly, RNA-seq reads were examined by a custom R script based on the ShortRead package to remove low-quality ($Q$ value <20) reads, adapters, and barcode

sequences. High-quality clean reads were assembled de novo into contigs by Trinity with strand-specific option "-SS_lib_type" set to "F" and "min_kmer_cov" set to 2. The resulting contigs were blasted against GenBank nonredundant, UniProt, and Arabidopsis protein databases with a cutoff $E$ value of 1e−5.

**5′ RLM-RACE.** To map the cmo-miRNA cleavage site of the *CmSPL* transcripts, a modified 5′ RLM-RACE procedure was used with the FirstChoice® RLM-RACE kit (Ambion)[61]. One μg total RNA was directly ligated to the 5′ adapter using the T4 RNA ligase without treatment with calf intestine alkaline phosphatase and tobacco acid pyrophosphatase. The ligated RNA was used to synthesize the cDNA using M-MLV reverse transcriptase according to the manufacturer's instructions (Ambion). PCR products were gel-purified and cloned into the pGEM T-Easy vector (Promega), and 10 clones were randomly selected and sequenced. The primers used in this assay are listed in Supplementary Table 1.

**ChIP assay.** For chrysanthemum, expanded leaves of *35S:CmNF-YB8-GFP* chrysanthemum plants 7 days after transplanting were used as materials. For *A. thaliana*, two-week-old *Super:CmNF-YB8-GFP A. thaliana* leaves were used. The plant tissues were fixed in formalin (final concentration, 1%) for 40 min. Fixed tissues were homogenized and the chromatin complexes were isolated and fragmented by sonication using a sonifier (Branson S-250D). The solubilized chromatin was immuno-precipitated overnight by 1 μl anti-GFP antibody (ab290, Abcam) with protein A-agarose (sc-2001, Santa Cruz Biotechnology). After washing, the beads were incubated at 65 °C for 6 h to reverse the cross-linking. The co-precipitated DNA was purified and analyzed using qRT-PCR. Primers used for the ChIP-qPCR are listed in Supplementary Table 1.

**Yeast one-hybrid assay.** For bait construction, cmo-MIR156 promoter fragments were amplified from chrysanthemum genomic DNA. A mutation was introduced into the CCAAT box from the cmo-MIR156 promoter fragment P9 using an asymmetric overlap extension PCR method[62]. The PCR products were then cloned into the EcoRI/SpeI sites of the pHIS2.1 vector. For prey construction, the coding region of CmNF-YB8 was amplified from chrysanthemum apical bud cDNA, and then cloned into the NdeI/XhoI sites of the pGADT7 vector. Yeast one-hybrid assays were conducted using the Matchmaker™ Gold Yeast One-Hybrid Library Screening System (Clontech). Briefly, transformants were grown on synthetic defined (SD)/-Trp-Leu plates to perform a spot assay. The transformants were then spotted onto SD/-Trp-leu-His, and SD/-Trp-leu-His + 3-amino-1,2,4-triazole (3-AT) plates.

**Dual-luciferase reporter assay in N. benthamiana.** To test the interaction between CmNF-YB8 and the cmo-MIR156 promoter in vivo, two pGreenII vectors were used: pGreenII 0800-LUC and pGreenII 0029 62-SK[63, 64]. The cmo-MIR156 promoter and a mutated version of the cmo-MIR156 promoter were inserted into the BamHI/NcoI sites of the pGreenII 0800-LUC vector at the 5′ end of the LUC gene. The CmNF-YB8 ORF was inserted into the BamHI/KpnI sites of the pGreenII 0029 62-SK vector. The pGreenII 0800-LUC vector carries a renilla luciferase (REN) gene under the control of the 35S promoter as a positive control.

To test cmo-miR156-mediated repression of the CmSPL target genes, potential cmo-miR156 cleavage sites from CmSPL3, 5, and 9 were inserted into the AvrII/AgeI sites of the pGreen-dualluc-ORF-sensor or pGreen-dualluc-3′UTR-sensor vectors[65]. A negative control (synonymous negative control, SNC) with disruption of CmSPL9 cleavage site while maintaining identical amino acid sequence was inserted into the AvrII/AgeI sites of the pGreen-dualluc-ORF-sensor vector. A 20 nt random sequence (Spacer) served as negative control for CmSPL3 and CmSPL5 was inserted into the AvrII/AgeI sites of the pGreen-dualluc-3′UTR-sensor vector. The cmo-miR156 overexpressor construct was derived from a pGreenII-based vector containing a CaMV 35S promoter, and the cmo-miR156a precursor[65]. The pGreen-dualluc-ORF-sensor and pGreen-dualluc-3′UTR-sensor vectors contain the REN gene under the control of the 35S promoter as a positive control.

All the constructs were transformed into *Agrobacterium tumefaciens* strain GV3101 harboring the pMP90 and pSoup plasmids. *Agrobacterium* cultures were inoculated in Luria-Bertani (LB) medium with selection antibiotics. The cultures were harvested by centrifugation at 4000 rpm for 10 min, and re-suspended in infiltration buffer (10 mM MgCl₂, 200 mM acetosyringone, 10 mM MES, pH 5.6) to an optical OD₆₀₀ of 0.7–1.0. *N. benthamiana* plants with 3–5 young leaves were used as materials. *Agrobacterium* culture mixtures (1:1) were infiltrated into *N. benthamiana* leaves using needleless syringes. After 3 days of infiltration, LUC and REN activities were measured using the dual-luciferase reporter assay reagents (Promega) and a GloMax 20/20 luminometer (Promega). The ratios of LUC and REN were expressed as activation or repression.

**Virus-based miRNA expression.** cmo-miR156 was ectopically expressed in chrysanthemum using a virus-based microRNA expression system[39]. A modified CaLCuV vector containing pre-cmo-miR156a (CaLCuV + miR156) was constructed and transformed into *A. tumefaciens* strain GV3101. The transformed *A. tumefaciens* cultures were inoculated in LB medium containing 30 mg l⁻¹ rifampicin, 50 μg ml⁻¹ kanamycin, and 50 μg ml⁻¹ gentamycin sulfate. The cultures were harvested by centrifugation at 4000 rpm for 10 min, and re-suspended in infiltration buffer (10 mM MgCl₂, 200 mM acetosyringone, 10 mM MES, pH 5.6) to

a final OD₆₀₀ of ~1.5. The cultures containing pCVB and CaLCuV (control), or pCVB and CaLCuV + miR156 were mixed in a 1:1 ratio, and placed at room temperature in the dark for 3–4 h before vacuum infiltration into chrysanthemum plantlets. The mock sample corresponded to the infiltration buffer alone. Chrysanthemum RNAi and WT plants, propagated in vitro, were immersed in infiltration buffer and infiltrated under a vacuum at 0.7 MPa for 3 min, and then washed in deionized water. After infiltration, the plants were transplanted into pots containing a mixture of 1:1 (v/v) peat and vermiculite and grown in a culture room at 23 ± 1 °C, 40% relative humidity, and 100 μmol m⁻² s⁻¹ illumination with fluorescent lamps, under a SD cycle (8 h light/16 h dark). Three independent experiments were performed with 40 plantlets in each experiment. Prior to the flowering time test, the plants were screened by PCR to determine the expression of cmo-miR156. cmo-miR156 overexpressing plants were then used for the flowering time test.

**Data availability.** RNA-seq data associated with this study has been deposited in NCBI SRA database under accession code PRJNA393366. The authors declare that all data supporting the finding of this study are available within the article and its Supplementary files or are available from the corresponding author on request.

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

## Acknowledgements

We thank Dr So Youn Won (Rural Development Administration, South Korea) for providing the sequence of the *cmo-MIR156* gene from his chrysanthemum genome sequencing data. We thank PlantScribe (www.plantscribe.com) for careful editing of this article. This work was supported by the National Natural Science Foundation of China (Grants 31372094, 31572157, and 31171990,).

## Author contributions

B.H., J.G., and C.M. conceived and designed the experiments; Q.W. performed most of the experiments; C.M. and C.-Z.J. provided technical support and conceptual advice and analyzed the data; Y.X., T.W., and L.Z. contributed to the ChIP assay and 5′ RLM-RACE; Y.X. and J.L. contributed to chrysanthemum transformation; T.W. and Y.C. contributed to miR156 cloning and functional analysis. C.M., B.H., and J.G. wrote the manuscript.

## Additional information

**Competing interests:** The authors declare no competing financial interests.

