## [Peer Review File · Nature Communications]

Reviewers' comments:

Reviewer #1 (Remarks to the Author):

In this manuscript, Wei et al., reports that CmNF-YB8 controls flowering time by regulating miR156. The data support by the transgenic approaches as well as biochemical experiments. As far as I know, this is first report on the up-stream regulator of MIR156 in plants. It is very interesting for the plant scientist to understand the regulatory mechanism of MIR156 and flowering time pathway. Then followings are my comments on this manuscript.

1. NF-YB as an up-stream regulator of MIR156: The ChIP data in this manuscript convince the binding of NF-YB in MIR156 promoter. However, the binding does not necessarily mean up-regulation. Transient assay for miR156 upregulation by a NF-YB might give better supporting information for this regulation. I guess the in vivo data is required.
2. Age-dependent flowering pathway: The author proposed that NF-YB is involved in the age dependent flowering pathway but not in the GA or photoperiod pathway. It will be better if the author discuss more why the other pathway is not involved by referring the previous reports.
3. Nomenclature of MIR156: In plants, if there are more than one precursors for the miRNA, they are named as MIRXXXa and MIR156b. So, instead of MIR156-1 and MIRT156-2, they should be named as MIR156a and MIR156b
4. Mapping of Cleavage sites: In Figure 6, the cleavage sites of SPL RNAs were represented. The major cleavage sites were located between 9th and 10th instead of the canonical 10th and 11th. This implies that the other miR156 variants, such as other members or isomiRs, might contribute the cleavage. This should be discussed in the manuscript. In addition, since SPL genes are RNA, "U" should be used instead of "T".

Minor

1. How TSSs of MIT155 were identified?
2. In table 1, what are the numbers mean in WT and RNAi? Are they RPKM values?

Reviewer #2 (Remarks to the Author):

Understanding the pathway that regulates age-dependent flowering is an exciting area of research because this pathway determines competency to flower, and with competency, an adult plant can respond to exogenous signals such as day-length, temperature etc. ensuring ecologically appropriate reproduction. A key component of this pathway, that appears to be evolutionarily conserved, is the post-transcriptional negative regulation of SPL transcription factors by members of the miRNA156 family. With age, levels of miRNA156 decline, resulting in a corresponding increase in SPL gene products, and therefore a transition from the juvenile to adult life history stage. Some regulators of miRNA156 have been identified in Arabidopsis, but how miRNA156 undergoes age-dependent downregulation is largely unknown.

Therefore, I find the results presented in Wei et al.'s manuscript very exciting.

Wei et al. use Chrysanthemum as a model, which is interesting because it is a short-day plant, while most SPL-related flowering time work is done in day neutral or long day plants. Wei et al. provide detailed and compelling evidence that in Chrysanthemum, a NF-Y transcription factor (CmNF-YB8) is involved in the direct regulation of Cm-miRNA156 and that Cm-miRNA156 in turn is a direct negative regulator of two SPL genes. NF-Y proteins are known to regulate chromatin epigenetic states. Therefore, this work will reach a broad audience of both plant biologists interested in the regulation of flowering time, as well as those interested in epigenetic control of gene expression states.

The result from this study are novel and well-substantiated by carefully presented data. However, I do

have a few recommendations that would improve the quality of the manuscript for Nature Communications.

1. Heterologous work in Arabidopsis.

The authors have put quite a bit of effort into undertaking heterologous experiments in Arabidopsis. CmNF-YB over-expression in Arabidopsis which prolongs the juvenile phase and delays flowering. And ChIP-PCR assays showing the CmNF-YB binds at the At-miRNA156 promoter. It is unclear to this reviewer what is substantially gained from these heterologous experiments, and I suggest that more appropriate would be testing the conservation of AtNF-YB(8/10) in miRNA156 regulation in Arabidopsis.

CmNF-YB8 overexpression in Arabidopsis prolongs the juvenile phase and delays flowering, but does AtNF-YB8(10) function similarly?

CmNF-YB8 binds at the At-miRNA156 promoter, but do AtNF-YB8(10) show a conserved pattern of involvement in At-miRNA156 regulation?

The authors argue that the data from these heterologous studies confirm that CmNF-YB8 functions in an aging pathway, but they do not provide any additional evidence beyond what was found in Chrysanthemum. The interesting question is whether the role of CmNF-YB8 is phylogenetically restricted (to Chrysanthemum and possibly relatives), or evolutionarily conserved (functioning similarly in Arabidopsis).

2. Assessment of CmSPL3/5/9 orthology.

The authors argue with little evidence to support the claim that the SPL transcripts they identified via their comparative RNAseq experiment correspond to SPL3, 5 and 9 in Arabidopsis. This may well be the case, but a phylogeny of the recovered CmSPL genes with Arabidopsis (and possibly other) SPL genes should be presented as a supplemental document.

3. Nicotiana transient expression assay.

CmSPL5:GFP appears to have similar signal reduction as CmSPL3:GFP and CmSPL9:GFP. The evidence is strong that Cm-miRNA156 cleaves CmSPL3 and CmSPL9, but perhaps the authors could either better explain the CmSPL5 result (the discrepancy between absence of binding site and decrease in GFP signal) or quantify the signal in order to demonstrate that CmSPL5 is significantly higher than CmSPL3/9.

Reviewer #3 (Remarks to the Author):

This study "The nuclear factor CmNF-YB8 controls the chrysanthemum flower aging pathway by regulating miR156" attempts to present that CmNF-YB8, a component of NF-Y complex in chrysanthemum, affects the age-related juvenile to adult vegetative transition by acting as a direct upstream component of miR156. CmNF-YB8 exhibited age-related decrease of its mRNA abundance as miR156 does. Knockdown of CmNF-YB8 by RNAi method led to earlier production of leaves with adult morphology and accelerated flower bud emergence that have correlation with the observation when the functions of miR156 were depleted in Arabidopsis. Consistently, miR156 abundance was decreased but higher accumulation of SPLs, the targets of miR156, was detected in the RNAi plants. The authors also demonstrated a direct binding of NF-YB8 proteins at specific motif in the promoter of miR156 precursor genomic locus in chrysanthemum.

miR156-SPLs is a module broadly conserved across plant lineages and generally believed to play roles

in the age-related developmental transition in plants. Abundance of miR156 decreases as plants become older and many genetic analyses visualized that the age-dependent downregulation of miR156 is crucial for the juvenile to adult vegetative transition in plants. Despite of its importance, molecular basis of the age-dependent miR156 downregulation is still elusive and a demanding issue in the field. So this study deals with an important biological question and the findings in the study can enhance our understanding on how miR156 levels are controlled in plants. However, the study employs plant materials that were propagated from cuttings of shoots presumably from mature plants and this experimental procedure can recruit some biological issues as described further in the review points. Additionally, only one RNAi transgene seems to be tried while the method possesses concerns on the specificity. Phenotypic analyses of CmNF-YB8 ox plants which were generated for ChIP-qPCR experiment are missing while those results are expected to be crucial to verify the main claims of the study. So I would recommend to invite the authors to revise their manuscript to address specific concerns before a final decision is reached. Below are the points to be addressed by the authors.

<Review points>

1. A major issue of the study is the way of how plant materials were prepared. In the study, chrysanthemum plants were propagated by regenerating plants from cuttings of shoots probably from mature plants. I wonder whether this type of plant culture system can be used to address any age-related developmental processes in the plants. For example, it should be critical to consider if plants regenerated from shoots of mature plants can undergo the age-related process again as like plants germinated from seeds. It should be better if the authors can present evidences demonstrating any developmental reprogramming in the regenerated plants at molecular levels (e.g. comparison of miR156 levels before, during and after regeneration) or use plants germinated from seeds in their study.
2. To address the roles of CmNF-YB8, the authors invited the RNAi method which has an issue on the specificity of the technique. I wonder if the authors tried only one RNAi transgene or examined more than two RNAi constructs which target independent regions of CmNF-YB8 mRNA. If they did try with only one RNAi transgene, there could be an issue that the observed phenotypes in the RNAi plants are caused by downregulation of unexpected non-specific target.
3. In Supplementary Figure 5, the authors examined mRNA levels of all members of the related NF-YB components by RT-qPCR and demonstrated that the RNAi transgene downregulates CmNF-YB8 alone among the family members. I wonder whether the authors can also present the mRNA levels of the related members in their RNA-seq results obtained from the RNAi plants. The results would be helpful to further support the specific downregulation of CmNF-YB8.
4. In Figure 5, the authors argue that the miR156-SPLs pathway in chrysanthemum is affected by the RNAi-mediated knockdown of CmNF-YB8 by demonstrating increased mRNA levels of SPLs and reduced miR156 abundance. It should be better to analyze and to present mRNA levels of another miR156-targeted SPLs in chrysanthemum to further support the authors' idea.
5. In Figure 5 and 7, the authors identified and examined the expression levels of cmo-miR156 precursor. Did the authors find only one miR156 precursor gene or are there additional genomic loci of miR156 precursors in chrysanthemum genome? It would be better if the authors clearly describe those information in the main text.
6. In Figure 7A, binding of CmNF-YB8 proteins at the promoter of cmo-miR156 precursor gene is shown by ChIP-qPCR analysis. To perform the analysis, the authors generated chrysanthemum plants overexpressing CmNF-YB8-GFP proteins. Then, I wonder if the authors observed any phenotypes in

the CmNF-YB8 ox plants that are inversed to those observed in CmNF-YB8 RNAi plants, such as delayed flowering, extended production of leaves with juvenile morphology, increased accumulation of miR156 so on. It would be worth to check these phenotypes to examine the main claims of the study.

7. In Figure 8, the authors claim that ectopic expression of miR156 delays floral bud emergence in CmNF-YB8 RNAi plants and suggest that miR156 is genetically downstream of CmNF-YB8. Were there any effects from miR156 ox on the age-related leaf morphogenesis as well?

<Editorial points>

8. Introduction, line39-42. The authors mentioned "such as the MADS-box genes, FLOWERING LOCUS T (FT), SUPPRESSOR OF OVEREXPRESSION OF CO1 (SOC1), and APETALA1 (AP1), and a plant specific transcription factor, LEAFY (LFY)". But FT is not a MADS-box protein. It should be better if the authors could check and present the information in a correct way.

9. Introduction, line 59. "Arabis alpine" is incorrect. "Arabis alpina" is the correct species name.

10. Results. It is unclear how and why the authors conceived to study biological roles of NF-YBs in the age-related developmental processes in chrysanthemum. It would be better if the authors could include their intellectual basis and some background information for the study.

11. Figure 1A. It should be better to include names of species from which the NF-YB8 homologues were characterized in Figure Legends.

12. Supplementary Figure 1. I would suggest the authors to include an information related to the nucleotide region in cDNA used for the RNAi of CmNF-YB8.

13. Discussion part of the manuscript sounds poor in principle. I suggest that the authors provide more enriched discussion for the results in the study.

We greatly appreciate the detailed and thorough reviews, which allowed us to clarify, expand on and improve our manuscript. We highlighted the changes by red in our revised manuscript.

Response to Reviewer #1

Question 1:

NF-YB as an up-stream regulator of MIR156: The ChIP data in this manuscript convince the binding of NF-YB in MIR156 promoter. However, the binding does not necessarily mean up-regulation. Transient assay for miR156 upregulation by a NF-YB might give better supporting information for this regulation. I guess the in vivo data is required.

Response:

Thank you for the helpful comment. We performed a dual-luciferase reporter assay to analyze the regulation of *cmo-MIR156* promoter activity by CmNF-YB8 *in vivo*. Our results showed that *Nicotiana benthamiana* leaf cells co-transformed with 35S:CmNF-YB8 and the promoter *cmo-MIR156:LUC* had significantly higher LUC activity compared with controls, indicating that the promoter activity of *cmo-MIR156* can be induced by interaction with CmNF-YB8. We added new data in Fig. 7e, and relevant descriptions in the 'Results' section of our revised manuscript (Line 220-225).

Question 2:

Age-dependent flowering pathway: The author proposed that NF-YB is involved in the age dependent flowering pathway but not in the GA or photoperiod pathway. It will be better if the author discuss more why the other pathway is not involved by referring the previous reports.

Response:

According to the suggestion of the reviewer, we have now expanded the 'Discussion' section (Line 263-288) of our revised manuscript. Briefly, the plants in the juvenile phase are thought to not be flowering competent, and will remain vegetative even when exposed to flowering stimulus cues, such as photoperiod or vernalization. The propensity of plants to flower and to initiate reproduction increases with age. Therefore, the basis of controlling age-related competence of flowering should be distinguishable from mechanism controlled by environmental responses, such as photoperiod, or by general endogenous pathways such as the GA pathway.

Previous studies of several plant species have demonstrated that *MIR156*, a key component of the aging pathway, regulates the transition from the juvenile to the adult vegetative phase. In addition, the expression of *MIR156* is affected by age, rather than other flowering cues, such as photoperiod or GA. In this study, we showed that CmNF-YB8 has a similar expression pattern to *MIR156*, as described above, and plays a role upstream of *MIR156* in chrysanthemum. Therefore, we consider it likely that CmNF-YB8 functions in regulating flowering time predominately through the aging pathway.

Question 3:

Nomenclature of MIR156: In plants, if there are more than one precursors for the miRNA, they are named as MIRXXXa and MIR156b. So, instead of MIR156-1 and MIRT156-2, they should be named as MIR156a and MIR156b.

Response:

We have re-named miR156 precursor-1 and -2 as miR156a and miR156b, respectively, in our revised manuscript.

Question 4:

Mapping of Cleavage sites: In Figure 6, the cleavage sites of SPL RNAs were represented. The major cleavage sites were located between 9th and 10th instead of the canonical 10th and 11th. This implies that the other miR156 variants, such as other members or isomiRs, might contribute the cleavage. This should be discussed in the manuscript. In addition, since SPL genes are RNA, "U" should be used instead of "T".

Response:

1) According to the suggestion of the reviewer, we added relevant content in the 'Discussion' section (Line 291-298) of our revised manuscript. Briefly, we made clear that although the cleavage site in *CmSPL9* was located between the canonical 10th and 11th nucleotide, the site in *CmSPL3* was between the 9th and 10th nucleotide. In addition, no site was identified in *CmSPL5*, but the expression of *CmSPL5* was altered by *CmNF-YB* silencing. This implied that other miR156 variants, such as other members or iso-miR156s, might contribute to the cleavage.

2) We changed T to U in Fig. 6a of the revised manuscript.

Question 5:

How TSSs of MIR156 were identified?

Response:

We isolated the full-length *cmo-MIR156* primary transcript and identified the TSS using the RLM-RACE approach, as previously described (Xie et al., 2005). This is stated in the 'Methods' section of our revised manuscript (Line 345-353).

Reference:

Xie, Z. et al. Expression of Arabidopsis MIRNA genes. *Plant Physiol* 138, 2145-54 (2005).

Question 6:

In table 1, what are the numbers mean in WT and RNAi? Are they RPKM values?

Response:

Yes, they are RPKM values, and we have stated this in Table 1 of our revised manuscript.

Response to Reviewer #2

Question 1:

Heterologous work in Arabidopsis.

The authors have put quite a bit of effort into undertaking heterologous experiments in Arabidopsis. CmNF-YB over-expression in Arabidopsis which prolongs the juvenile phase and delays flowering. And ChIP-PCR assays showing the CmNF-YB binds at the At-miRNA156 promoter. It is unclear to this reviewer what is substantially gained from these heterologous experiments, and I suggest that more appropriate would be testing the conservation of AtNF-YB (8/10) in miRNA156 regulation in Arabidopsis.

CmNF-YB8 overexpression in Arabidopsis prolongs the juvenile phase and delays flowering, but does AtNF-YB8 (10) function similarly?

CmNF-YB8 binds at the At-miRNA156 promoter, but do AtNF-YB8 (10) show a conserved pattern of involvement in At-miRNA156 regulation?

The authors argue that the data from these heterologous studies confirm that CmNF-YB8 functions in an aging pathway, but they do not provide any additional evidence beyond what was found in Chrysanthemum. The interesting question is whether the role of CmNF-YB8 is phylogenetically restricted (to Chrysanthemum and possibly relatives), or evolutionarily conserved (functioning similarly in Arabidopsis).

Response:

Thanks for the valuable comment. To address this question:

1) We obtained one Arabidopsis *NF-YB8* t-DNA insertion mutant (Salk_108199) and one *NF-YB10* t-DNA insertion mutant (Salk_135018) from the Arabidopsis Biological Resource Center (Fig. 1a below). However, genotyping analysis showed that the insertion site of Salk_108199 is located at the 3'UTR of *NF-YB8*, and the insertion site of Salk_135018 is located in front of the 5'UTR of *NF-YB10* (Fig. 1a and b below). The expression of *NF-YB8* and *NF-YB10* was not significantly altered in these mutants (Fig. 1c below). Moreover, we did not observe any difference in phenotype between WT and the mutants (Fig. 1d below). Therefore, we did not pursue using these mutants to further test the functions of *NF-YB8* and *NF-YB10*.

Figure 1. Phenotype of Arabidopsis *nf-yb8* and *nf-yb10* mutants. (a) Transfer DNA insertion sites in *NF-YB8* (AT2G37060) and *NF-YB10* (AT3G53340). Black bars indicate exons, and white bars indicate introns. LP, RP, and LB indicate the location of primers used for genotyping. F and R indicate the location of primers used for gene expression analysis. (b) DNA insertion sites were determined by PCR analysis. (c) The expression of *NF-YB8* and *NF-YB10* genes in wild-type (WT) and mutants was determined by RT-PCR. (d) Phenotype of *nf-yb8* and *nf-yb10* mutants.

2) We tested the expression of *NF-YB8* and *NF-YB10* during the growth of *Arabidopsis* using *ProNF-YB8:GUS* and *ProNF-YB10:GUS* plants. The *Arabidopsis ProNF-YB8:GUS* (CS67026) and *ProNF-YB10:GUS* (CS67028) (Siefers et al., 2009) lines were obtained from the *Arabidopsis* Biological Resource Center. GUS staining analysis showed that *NF-YB8* was expressed in the vascular tissue of young leaves, and a *NF-YB10* signal was not detected (Fig. 2 below).

Figure 2. *NF-YB8* and *NF-YB10* expression patterns during the *Arabidopsis* life cycle.

3) We tested the regulation by *NF-YB8* and *NF-YB10* of the promoter activity of *MIR156A* by use of a dual-luciferase reporter assay in *Nicotiana benthamiana* leaves. The results showed that *Nicotiana benthamiana* leaf cells co-transformed with *35S:NF-YB10* and the promoter of *MIR156A:LUC* (*Pro-miR156a*) had higher LUC activity than cells transformed with *NF-YB10* or *Pro-miR156* alone (Fig. 3 below).

Figure 3. Interaction between *NF-YB8* or *NF-YB10* with the *MIR156A* promoter shown by use of a dual-luciferase reporter assay in *Nicotiana benthamiana* leaves. A *MIR156A* promoter fragment spanning from 0--1034 bp was used in this assay. LUC vectors contain the REN gene under the control of the 35S promoter as a positive control. *Nicotiana benthamiana* leaves were infiltrated with the samples, and LUC and REN activities were assayed 3 d after infiltration. The ratio of LUC/REN of the empty vector (SK) co-transformed with the promoter of *MIR156A:LUC* (*Pro-miR156a*) vector was used as the calibrator (set as 1). Three independent experiments were performed and error bars indicate standard deviation.

4) We created *35S:NF-YB8* and *35S:NF-YB10* *Arabidopsis* plants. We observed that *35S:NF-YB10* plants (T1) had smaller and rounder leaves compared to WT and *35S:NF-YB8*

plants (Fig. 4 below). In addition, we observed a delay in flowering time in *35S:NF-YB10* plants compared to WT and *35S:NF-YB8* plants (Fig. 4 below).

Based on our preliminary results above, we consider that the role of *CmNF-YB8* in the aging pathway is probably evolutionarily conserved in chrysanthemum and *Arabidopsis*. But further investigation should be done to confirm the function of NF-YB10/NF-YB8 in aging pathway in *Arabidopsis*.

Figure 4. Phenotypes of *Arabidopsis thaliana* *35S:NF-YB8* and *35S:NF-YB10* lines.

Reference:

Siefers, N., Dang, K.K., Kumimoto, R.W., Bynum, W.E., Tayrose, G., and Holt, B.F. 2009. Tissue-specific expression patterns of *Arabidopsis* NF-Y transcription factors suggest potential for extensive combinatorial complexity. *Plant Physiology* 149:625-641.

Question 2:

Assessment of CmSPL3/5/9 orthology.

The authors argue with little evidence to support the claim that the SPL transcripts they identified via their comparative RNAseq experiment correspond to SPL3, 5 and 9 in Arabidopsis. This may well be the case, but a phylogeny of the recovered CmSPL genes with Arabidopsis (and possibly other) SPL genes should be presented as a supplemental document.

Response:

We now present a phylogenetic analysis of CmSPLs with AtSPLs in supplemental Fig. 8 of the revised manuscript.

Question 3:

Nicotiana transient expression assay.

CmSPL5:GFP appears to have similar signal reduction as CmSPL3:GFP and CmSPL9:GFP. The evidence is strong that Cm-miRNA156 cleaves CmSPL3 and CmSPL9, but perhaps the authors could either better explain the CmSPL5 result (the discrepancy between absence of binding site and decrease in GFP signal) or quantify the signal in order to demonstrate that CmSPL5 is significantly higher than CmSPL3/9.

Response:

To address this question, we performed a dual-luciferase-based miRNA sensor assay (Liu et al., 2014) to quantitatively evaluate miR156 cleaving of *CmSPLs*. The results showed that co-expression of 35S:pre-cmo-miR156a with either of the *CmSPL3* or *CmSPL9* cleavage sites fused to the *Firefly luciferase (LUC)* gene led to reduced LUC activity in *Nicotiana benthamiana*. However, this was not observed when 35S:pre-cmo-miR156a was co-expressed with the *CmSPL5* cleavage site fused to the *LUC* gene. We added new data in Fig. 6c and relevant description in the 'Results' section of our revised manuscript (Line 192-203).

Reference:

Liu Q, Wang F, Axtell M.J. (2014) Analysis of complementarity requirements for plant microRNA targeting using a *Nicotiana benthamiana* quantitative transient assay. *Plant Cell*, 26:741-753.

Response to Reviewer #3

Question 1:

A major issue of the study is the way of how plant materials were prepared. In the study, chrysanthemum plants were propagated by regenerating plants from cuttings of shoots probably from mature plants. I wonder whether this type of plant culture system can be used to address any age-related developmental processes in the plants. For example, it should be critical to consider if plants regenerated from shoots of mature plants can undergo the age-related process again as like plants germinated from seeds. It should be better if the authors can present evidences demonstrating any developmental reprogramming in the regenerated plants at molecular levels (e.g. comparison of *miR156* levels before, during and after regeneration) or use plants germinated from seeds in their study.

Response:

Thank you for the valuable feedback. Chrysanthemum is heterozygous and offspring propagated by seeds always segregate. Therefore, micro-cutting propagation as a way of asexual reproduction is commonly used to facilitate the genetically conserved propagation of relevant cultivars.

The occurrence of rejuvenation during *in vitro* micropropagation has been reported in several plant species, such as grape (Mullins et al., 1979), deciduous azaleas (Economou and Read, 1986), apple (Sriskandarajah et al., 1982), elderberry (Hrib et al., 1980), raspberry (Broome and Zimmerman, 1978), and birch (Brand and Lineberger, 1992). However, in chrysanthemum, no reports of rejuvenation by *in vitro* micropropagation are available. In this study, we found that juvenile leaves regenerated from micro-cutting propagated chrysanthemum plantlets (Fig. 3 in the manuscript), indicating the rejuvenation does occur during micro-cutting propagation.

In addition, according to the suggestion of the reviewer, we examined the expression of *cmo-MIR156* in chrysanthemum before and after micro-cutting propagation, by qRT-PCR. We observed that the expression of *cmo-MIR156* increased in leaves of propagated young plants (two-leaf old regenerated plants) compared to in the leaves before propagation (mother plant), and declined in the leaves of the propagated mature plants (ten-leaf old regenerated plants, Figure below).

Figure. Expression analysis of *cmo-MIR156* in chrysanthemum plants regenerated from shoots of mature plants. The leaves were collected from mature plants used for propagation (mother plant), and regenerated plants with two leaves and ten leaves. Three independent experiments were performed and error bars indicate standard deviation. Asterisks indicate significant differences according to a Student's t-test (** $P < 0.01$).

Reference:

- Brand, M.H., and Lineberger R.D. 1992. *In vitro* rejuvenation of *Betula* (Betulaceae): morphological evaluation. *American Journal of Botany* 79:618-625.
- Broome, O.C., and Zimmerman, R.H. 1978. *In vitro* propagation of blackberry. *HortScience* 13:151-153.
- Economou, A.S., and Read, P.E. 1986. Microcutting production from sequential reculturing of hardy deciduous azalea shoot tips. *HortScience* 21:137-139.
- Hrib, J.H., Chaturvedi, C., and Dobry, J. 1980 Development of shoot tips of *Sambucus* sp. Grown *in vitro*. *Phytomorphology* 30:266-270.

Mullins, M.G., Nair, Y., and Sampet, P. 1979. Rejuvenation in vitro: induction of juvenile characters in an adult clone of *Vitis vinifera* L. *Annals of Botany* 44:623-627.

Sriskandarajah, D., Mullins M.G., and Nair, Y. 1982. Induction of adventitious rooting in vitro in difficult-to-propagate cultivars of apple. *Plant Science Letters* 24:1-9.

Question 2:

To address the roles of CmNF-YB8, the authors invited the RNAi method which has an issue on the specificity of the technique. I wonder if the authors tried only one RNAi transgene or examined more than two RNAi constructs which target independent regions of CmNF-YB8 mRNA. If they did try with only one RNAi transgene, there could be an issue that the observed phenotypes in the RNAi plants are caused by downregulation of unexpected non-specific target.

Response:

We agree that it is better to examine function of *CmNF-YB8* using more than two RNAi constructs that target independent regions of *CmNF-YB8* mRNA. In this study, we used only one RNAi transgene; however, we tested the expression of all the members of the CmNF-YB family in the *CmNF-YB8*-RNAi chrysanthemum plants (supplementary Fig. 5 in the manuscript). We determined that with the exception of *CmNF-YB8*, none of the CmNF-YB family genes exhibited reduced expression levels in *CmNF-YB8*-RNAi plants compared to WT plants, thereby confirming the specificity of the gene silencing.

Question 3:

In Supplementary Figure 5, the authors examined mRNA levels of all members of the related NF-YB components by RT-qPCR and demonstrated that the RNAi transgene downregulates CmNF-YB8 alone among the family members. I wonder whether the authors can also present the mRNA levels of the related members in their RNA-seq results obtained from the RNAi plants. The results would be helpful to further support the specific downregulation of CmNF-YB8.

Response:

As the reviewer suggested, we identified all the members of the CmNF-YB family in our RNA-seq data, and found that the transcript levels of the CmNF-YB members, except *CmNF-YB8*, were not changed in *CmNF-YB8*-RNAi chrysanthemum plants (Table below).

Table. RNA-seq data of all the members of NF-YB family in chrysanthemum.

GeneID	Annotation	WT	RNAi	RNAi/WT fold change
UN126171	NF-YB1	132.109	145.505	1.10
UN128733	NF-YB2	0	0	—
UN001980	NF-YB4	22.8323	19.6526	0.86
UN029265	NF-YB6	7.34897	6.73074	0.92
UN075928	NF-YB18	63.1128	69.934	1.11
UN062189	NF-YB5	1.94983	1.37527	0.71
UN088090	NF-YB8	58.7191	23.49	0.40
UN102331	NF-YB5-like	0	0	—

Question 4:

In Figure 5, the authors argue that the miR156-SPLs pathway in chrysanthemum is affected by the RNAi-mediated knockdown of CmNF-YB8 by demonstrating increased mRNA levels of SPLs and reduced miR156 abundance. It should be better to analyze and to present mRNA levels of another miR156-targeted SPLs in chrysanthemum to further support the authors' idea.

Response:

According to the suggestion of the reviewer, we screened our RNA-seq data and found 4 other SPL genes: CmSPL2, CmSPL7, CmSPL12 and CmSPL14, of which the expression showed no significant difference between WT and the CmNF-YB8-RNAi plants. In addition, we also measured the transcript abundance of the four CmSPL genes in the CmNF-YB8-RNAi chrysanthemum plants by qRT-PCR (Figure below), and observed that the patterns of expression were generally consistent with the RNA-seq data.

Figure. Expression of CmSPL genes in CmNF-YB8-RNAi chrysanthemum plants. qRT-PCR was performed to evaluate the expression of each gene. UBIQUITIN was used as the internal control gene. Three independent experiments were performed and error bars indicate standard deviation.

Question 5:

In Figure 5 and 7, the authors identified and examined the expression levels of cmo-miR156 precursor. Did the authors find only one miR156 precursor gene or are there additional genomic loci of miR156 precursors in chrysanthemum genome? It would be better if the authors clearly describe those information in the main text.

Response:

Chrysanthemum is a hexaploid with a large genome. Moreover, to date, no whole-genome information is available. Dr. So Youn Won from the Rural Development Administration of South Korea generously provided the sequence of the cmo-MIR156 gene from his unpublished chrysanthemum genome sequencing data. We only found one MIR156 gene in his data. We describe this in the 'Methods' section of our revised manuscript (line 354-355).

Question 6:

In Figure 7A, binding of CmNF-YB8 proteins at the promoter of cmo-miR156 precursor gene is shown by ChIP-qPCR analysis. To perform the analysis, the authors generated chrysanthemum plants overexpressing CmNF-YB8-GFP proteins. Then, I wonder if the authors observed any phenotypes in the CmNF-YB8 ox plants that are inverted to those observed in CmNF-YB8 RNAi plants, such as delayed flowering, extended production of leaves with juvenile morphology, increased accumulation of miR156 so on. It would be worth to check these phenotypes to examine the main claims of the study.

Response:

Based on this suggestion, we analyzed the flowering time of *CmNF-YB8-OX* chrysanthemum plants, and showed that their flowering time was slightly delayed compared with WT plants. We added the new data in supplementary Fig. 6 and the relevant description in the 'Results' section of our revised manuscript (Line 142-147).

Question 7:

In Figure 8, the authors claim that ectopic expression of miR156 delays floral bud emergence in CmNF-YB8 RNAi plants and suggest that miR156 is genetically downstream of CmNF-YB8. Were there any effects from miR156 ox on the age-related leaf morphogenesis as well?

Response:

Thanks for the valuable comment. Since we infiltrated chrysanthemum plants that already had juvenile leaves, we could not assess the effect of virus-induced overexpression of cmo-miR156 on age-related leaf morphogenesis, especially on the number of juvenile leaves.

<Editorial points>

Question 8:

Introduction, line39-42. The authors mentioned “such as the MADS-box genes, FLOWERING LOCUS T (FT), SUPPRESSOR OF OVEREXPRESSION OF CO1 (SOC1), and APETALA1 (API), and a plant specific transcription factor, LEAFY (LFY)”. But FT is not a MADS-box protein. It should be better if the authors could check and present the information in a correct way.

Response:

We apologize for this mistake. We revised this sentence to ‘These flowering pathways converge on a common set of downstream flowering time integrators, such as *FLOWERING LOCUS T (FT)*, *SUPPRESSOR OF OVEREXPRESSION OF CO1 (SOC1)*, and *APETALA1 (API)*, and a plant specific transcription factor, *LEAFY (LFY)*.’

Question 9:

Introduction, line 59. “Arabis alpine” is incorrect. “Arabis alpina” is the correct species name.

Response:

Now corrected.

Question 10:

Results. It is unclear how and why the authors conceived to study biological roles of NF-YBs in the age-related developmental processes in chrysanthemum. It would be better if the authors could include their intellectual basis and some background information for the study.

Response:

Studies in our lab focus on the mechanisms of abiotic stress tolerance and flowering in chrysanthemum. We firstly explored the functions of chrysanthemum *NF-Y* members in abiotic stress tolerance, since several previous studies have demonstrated that *NF-Y* genes have functions in abiotic stress tolerance. We found that *CmNF-YB8* might play a role in drought stress tolerance. In addition, we observed that *CmNF-YB8-RNAi* chrysanthemum plants had altered flowering time compared to WT plants. Therefore, we explored the function of *CmNF-YB8* in flowering. Further investigation indicated that *CmNF-YB8* is not involved in photoperiod, vernalization, or GA flowering pathways, but involved in aging pathway.

Question 11:

Figure 1A. It should be better to include names of species from which the NF-YB8 homologues were characterized in Figure Legends.

Response:

According to the suggestion of the reviewer, we added plant species information of the NF-YB8 homologs to the legend of Fig. 1a.

Question 12:

Supplementary Figure 1. I would suggest the authors to include an information related to the nucleotide region in cDNA used for the RNAi of CmNF-YB8.

Response:

We have now added the information of the nucleotide region used for the RNAi of CmNF-YB8 to supplementary Fig. 1 of our revised manuscript.

Question 13:

Discussion part of the manuscript sounds poor in principle. I suggest that the authors provide more enriched discussion for the results in the study.

Response:

As suggested, we have expanded and improved the 'Discussion, also incorporating comments from the other two reviewers.

REVIEWERS' COMMENTS:

Reviewer #1 (Remarks to the Author):

In this revised manuscript, the authors well-addressed my previous comments. Especially, the transient expression of CmNF-YB8 in tobacco leaves proved the proof of the role of CmNF-YB8 as an upstream regulator of miR156. However, I have one question about this experiment. It has been known that NF-YB interacts with NF-YA and NF-YC as a heterodimer to regulate downstream target genes. I am wondering how over-expression of one of the NF-Y factor, NF-YB8, could affect the up-regulation of miR156 in tobacco leaves. I guess tobacco NF-YA and NF-YB interacted with CmNF-YB proteins. Please discuss this hypothesis in the discussion and provide the age of tobacco leaves for this experiment.

The second comment is about the role of NF-YB8 in the age-dependent flowering pathway. As the authors described in the introduction, several NF-Y genes in Arabidopsis are involved in the photoperiod pathway and GA pathway by interacting with CO proteins and relationship with DELLA proteins, respectively. I expected discussion about the new role of NF-YB gene in the age-dependent flowering pathway by comparing the other NF-Y genes in the other pathway. Please discuss more this issue in the discussion.

Reviewer #2 (Remarks to the Author):

In the revised version of this manuscript, the authors provide clear and substantiated evidence that a nuclear factor, CmNF-YB8, is an upstream regulator of the mir156/SPL age-dependent flowering pathway in Chrysanthemum. These results open new avenues of research for those interested in the genetic control of flowering time, especially in emerging model species. Concerns I raised in the earlier version of this manuscript have been addressed in this revised version.

Reviewer #3 (Remarks to the Author):

The major points raised in my review are addressed sufficiently in the revised works of the authors. There is still a little concern on the RNAi plant material but many independent results in the study consistently support the authors' main claim. So I pleasantly recommend the publication of the study on Nature Communications.

Minor suggestion

Many of works in the rebuttal letters have not been adopted in the revised manuscript. I suggest that the authors revisit their manuscript to include these results reflecting the points from reviewers.

Response to Reviewer #1

Comment 1:

In this revised manuscript, the authors well-addressed my previous comments. Especially, the transient expression of CmNF-YB8 in tobacco leaves proved the proof of the role of CmNF-YB8 as an upstream regulator of miR156. However, I have one question about this experiment. It has been known that NF-YB interacts with NF-YA and NF-YC as a heterodimer to regulate downstream target genes. I am wondering how over-expression of one of the NF-Y factor, NF-YB8, could affect the up-regulation of miR156 in tobacco leaves. I guess tobacco NF-YA and NF-YB interacted with CmNF-YB proteins. Please discuss this hypothesis in the discussion and provide the age of tobacco leaves for this experiment.

Response:

According to the comment of the reviewer, we added the relevant content in the 'Discussion' section (Lines 313-322).

Regarding the age of tobacco leaves, we used *N. benthamiana* plants with 3-5 young leaves in this assay. We stated it in the 'Methods' section of our revised manuscript (Lines 518-519).

Comment 2:

The second comment is about the role of NF-YB8 in the age-dependent flowering pathway. As the authors described in the introduction, several NF-Y genes in Arabidopsis are involved in the photoperiod pathway and GA pathway by interacting with CO proteins and relationship with DELLA proteins, respectively. I expected discussion about the new role of NF-YB gene in the age-dependent flowering pathway by comparing the other NF-Y genes in the other pathway. Please discuss more this issue in the discussion.

Response:

Thanks for the valuable suggestion of reviewer. We added relevant content in the 'Discussion' section of our revised manuscript (Lines 331-350). Briefly, in *A. thaliana*, NF-YB2 was discovered can associate with the CCAAT box of *FT* and *SOC1* promoters in both photoperiod and GA flowering pathway. Here, we found a new function of NF-YB subunit members in flowering time. It suggests broad-spectrum functions of NF-YB subunit members in regulation of flowering time in response to environmental cues and internal signals. In addition, the NF-Y complex has been demonstrated to be an epigenetic regulator in flowering pathway, and it is known that epigenetic regulation of miR156 expression occurs during the vegetative phase change. Therefore, we speculated NF-YB8 may be also involved in epigenetic regulation of miR156 expression.

Response to Reviewer #2

Comments:

In the revised version of this manuscript, the authors provide clear and substantiated evidence that a nuclear factor, CmNF-YB8, is an upstream regulator of the mir156/SPL age-dependent flowering pathway in Chrysanthemum. These results open new avenues of research for those

interested in the genetic control of flowering time, especially in emerging model species. Concerns I raised in the earlier version of this manuscript have been addressed in this revised version.

Response:

Thanks for the comments.

Response to Reviewer #3

Comments:

The major points raised in my review are addressed sufficiently in the revised works of the authors. There is still a little concern on the RNAi plant material but many independent results in the study consistently support the authors' main claim. So I pleasantly recommend the publication of the study on Nature Communications.

Minor suggestion

I suggest that the authors revisit their manuscript to include these results reflecting the points from reviewers.

Response:

Thanks for the valuable suggestion of reviewer. Nevertheless, we prefer organize our manuscript in the present way without adding the works in the rebuttal letters, because we have agreed with publishing the reviewer comments and our rebuttal letter.